# Zero-Flow Encoders

**Yakun Wang** [* 1]   **Leyang Wang** [* 2]   **Song Liu** [1]   **Taiji Suzuki** [2 3]

## Abstract

Flow-based methods have achieved significant success in various generative modeling tasks, capturing nuanced details within complex data distributions. However, few existing works have exploited this unique capability to resolve fine-grained structural details beyond generation tasks. This paper presents a flow-inspired framework for representation learning. First, we demonstrate that a rectified flow trained using independent coupling is zero everywhere at $t = 0.5$ if and only if the source and target distributions are identical. We term this property the *zero-flow criterion*. Second, we show that this criterion can certify conditional independence, thereby extracting *sufficient information* from the data. Third, we translate this criterion into a tractable, simulation-free loss function that enables learning amortized Markov blankets in graphical models and latent representations in self-supervised learning tasks. Experiments on both simulated and real-world datasets demonstrate the effectiveness of our approach. The code reproducing our experiments can be found at: https://github.com/probabilityFLOW/zfe.

## 1. Introduction

In recent years, continuous-time flow methods, such as diffusion models (Song & Ermon, 2019; Ho et al., 2020; Song et al., 2021a;b; Karras et al., 2022) and flow matching (Lipman et al., 2023; Liu et al., 2023), have been successfully applied to a broad range of generative modeling tasks, including image synthesis (Dhariwal & Nichol, 2021; Rombach et al., 2022), time-series forecasting (Tashiro et al., 2021), and simulation-based inference (Geffner et al., 2023;

---
[*]Equal contribution [1]School of Mathematics, University of Bristol, Bristol, UK [2]Center for Advanced Intelligence Project, RIKEN, Tokyo, Japan [3]Department of Mathematical Informatics, the University of Tokyo, Tokyo, Japan. Correspondence to: Song Liu <song.liu@bristol.ac.uk>.

*Proceedings of the 43rd International Conference on Machine Learning*, Seoul, South Korea. PMLR 306, 2026. Copyright 2026 by the author(s).

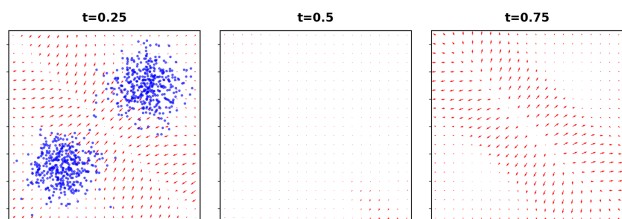

*Figure 1.* "Zero-flow" phenomenon. Rectified flow trained on identical distributions (a Gaussian mixture, blue points) with independent coupling. At $t = 0.5$, the flow becomes stationary almost everywhere. An explanation of this phenomenon can be found in Section 3, and additional visualizations can be found in Figure 9.

Wildberger et al., 2023; Sharrock et al., 2024). These methods learn a time-dependent velocity field that defines a flow, progressively transporting samples from a simple source distribution toward a complex target distribution over a continuous time interval. The widespread success of these approaches demonstrates their versatility across diverse datasets and their capability to capture the nuanced structural details of complex distributions.

Given the success of flow-based methods, it is natural to wonder whether they can be extended to other unsupervised tasks that often require flexible models to capture the subtle details of complex data distributions.

Recent works have sought to extend flow-based methods beyond generative tasks. For example, Li et al.; Beizaee et al. transport a data distribution to a Gaussian distribution for outlier detection. He et al. also transports a conditional distribution to an unconditional Gaussian distribution and performs conditional independence tests. Examples also arise in reinforcement learning, where Park et al. uses flow-matching policies to parametrize action distributions for Q-learning. Applying flow matching to non-generative tasks is a rapidly growing research area.

In this paper, we address a classic unsupervised learning task: extracting patterns and representations from data (Radford et al., 2021; Uelwer et al., 2023). We propose a flow model-inspired framework for these problems. First, we observe a phenomenon which we term *zero-flow*: a rectified flow with independent coupling vanishes at the midpoint ($t = 0.5$) if and only if the source and target distributions are identical (see Figure 1 and Appendix A for a comprehen-

sive elucidation). Second, we demonstrate that this property serves as a valid test for the equality of two conditional distributions. Third, we develop this criterion into a tractable least-squares loss function that learns a *sufficient encoding* by enforcing conditional independence.

We consider two applications of the proposed *zero-flow encoder*: Learning amortized Markov blankets and self-supervised representation learning. Given a set of target features, identifying its Markov blanket amounts to selecting a small subset of the remaining features conditioned on which, the other features are independent of the target features. The learned subsets often offer new insights into the dataset. Many existing methods assume a parametric density model for the data distribution and estimate it using a lasso-based subset selector (Meinshausen & Bühlmann, 2006; Banerjee et al., 2008; Nguyen & Basu, 2026). We demonstrate that our zero-flow encoder efficiently identifies Markov blankets in a non-parametric manner. Moreover, by employing an amortized encoding strategy, our method supports arbitrary target features, enabling flexible inference for target features not encountered during training.

Self-supervised representation learning aims to extract key global information without using any labels. Instead, they rely on creating supervisory information from the dataset itself by learning invariant or consistent information across different variations of the same data. The essence of these tasks can again be boiled down to enforcing conditional independence among different transformations (views) of the same data, given the global, semantic information. We show that this information can be accurately extracted using our zero-flow encoder. Existing methods for learning representations have achieved strong empirical performance by optimizing surrogate criteria such as maximizing the mutual information or pointwise mutual information between views (Bachman et al., 2019; Chen et al., 2020; Uesaka et al., 2025). However, these objectives suffer from "taking shortcuts", due to the greediness of mutual information maximization. They are easily affected by random, superficial patterns (such as a watermark), fails to identify high-level semantic features. We summarize our contributions.

- We identify the zero-flow condition in rectified flow, and mathematically prove that it holds if and only if the source and target distributions are perfectly aligned, in both unconditional and conditional cases.

- We utilize this condition to learn an encoder that extracts *sufficient information* for predicting target features, *without any explicit parametric assumption*.

- We leverage this criterion for learning Markov blankets in a graphical model, where the encoder selects a subset of features to predict target variables, offering insights into a high-dimensional dataset.

- We utilize this criterion for self-supervised learning, and show the learned encoder does not suffer from the "shortcut problem".

## 2. Background

We use capital letters, such as $X$ and $Y$, to represent random variables and lowercase boldface letters, such as $\mathbf{x}$ and $\mathbf{y}$, to represent vectors. Given a binary mask vector $\mathbf{m} \in \{0,1\}^d$, and $X \in \mathbb{R}^d$, $X_{\mathbf{m}}$ stands for the sub-vector of $X$ whose elements corresponds to ones in $\mathbf{m}$. $X_{-\mathbf{m}}$ represents the rest of the vector. The gradient of a function $f(\mathbf{x})$ is $\nabla f(\mathbf{x}) := [\partial_{x_1} f(\mathbf{x}), \ldots, \partial_{x_d} f(\mathbf{x})]^{\top}$. $p_X(\mathbf{x})$ is a density function of $X$ evaluated at a point $\mathbf{x}$. $\|\cdot\|$ represents $\mathcal{L}_2$ norm. $\circ$ denotes the elementwise product.

In this paper, we aim to design a flow-inspired criterion for extracting summary information from the dataset. For this reason, we briefly review a few related research areas.

### 2.1. Flow-based Generative Models

Our methodology is inspired by flow-based generative models. Given empirical observations from two distributions $X \sim p_X$ and $X' \sim p_{X'}$ on $\mathbb{R}^d$, the objective of flow-based generative model is to learn a deterministic transport map $T : \mathbb{R}^d \to \mathbb{R}^d$ such that the pushforward of the source distribution matches the target distribution: if $Z \sim p_X$, then $Z' := T(Z)$ satisfies $Z' \sim p_{X'}$.

Earlier works, such as continuous normalizing flows (CNFs) (Tabak & Vanden-Eijnden, 2010; Chen et al., 2018), parameterize $T$ as the solution to an Ordinary Differential Equation (ODE) of the form $\mathrm{d}Z(t) = \mathbf{v}_t(Z(t)) \, \mathrm{d}t$, where the velocity field $\mathbf{v}_t$ is defined by a time-dependent neural network. More recent generalizations have extended this concept to SDE (Sohl-Dickstein et al., 2015; Song & Ermon, 2019; Ho et al., 2020; Song et al., 2021b), in which neural networks approximate the drift term and numerical solvers are employed for inference-time simulation.

Rectified Flow (Liu et al., 2023) learns a deterministic transport map from paired samples $(X, X')$. By constructing an interpolation path $X_t = tX' + (1-t)X$, the method learns a velocity field $\mathbf{v}_t$ by minimizing the squared error against the straight-line direction $(X' - X)$. Initially, $X$ and $X'$ are drawn independently. However, Rectified Flow introduces a recursive Reflow procedure to straighten ODE trajectories. In this paper, when we refer to Rectified Flow, we only mean the initial flow training with independent pairs $(X, X')$. We do not consider Reflow in our algorithm.

### 2.2. Markov Blankets and Undirected Graphical Models

One way to summarize our dataset is to identify a small subset of features that are sufficient to predict the targeting

features. In this section, we briefly review such sufficiency through the lens of undirected graphical models and Markov blankets.

An undirected graphical model is defined as an undirected graph $G = (E, V)$, where $V \in \{1, \ldots, d\}$ are indices of a multivariate random variable $Z$. $G$ encodes the conditional independence of $Z$. See, e.g., Section 4 in Koller & Friedman (2009) for more details.

Given a binary partition of the graph, indicated by a mask $\mathbf{m} \in \{0,1\}^d$. The Markov blanket of the target feature $Z_{\mathbf{m}}$, denoted by $\mathcal{M}(Z_{\mathbf{m}})$, is defined as the minimal subset $\mathcal{M}(Z_{\mathbf{m}}) \subseteq Z_{-\mathbf{m}}$ such that

$$Z_{\mathbf{m}} \perp\!\!\!\perp Z_{-\mathbf{m}} \mid \mathcal{M}(Z_{\mathbf{m}}),$$

which is equivalent to the identity

$$p_{Z_{\mathbf{m}}|Z_{-\mathbf{m}}} = p_{Z_{\mathbf{m}}|\mathcal{M}(Z_{\mathbf{m}})}.$$

In general terms, $\mathcal{M}(Z_{\mathbf{m}})$ is a minimal subset of $Z_{-\mathbf{m}}$ that is required to achieve the best prediction of $Z_{\mathbf{m}}$.

Identifying the Markov blankets for target features provides a summary of the dataset and has been the key principle for learning sparse graphical models (Meinshausen & Bühlmann, 2006; Ravikumar et al., 2010; Yang et al., 2012; 2015; Meng et al., 2021). For a single variable $Z_i$, $\mathcal{M}(Z_i)$ is essentially its neighbours in $G$. By rotating the choice of $Z_i$, we can identify all edges in the graph. In other applications, the choice of the target features $Z_{\mathbf{m}}$ may depend on downstream applications. For example, clinicians may be interested in identifying genes associated with a set of symptoms, which may be unknown during training.

In this work, we propose a flow-inspired, non-parametric, amortized Markov blanket selector that enables inference for the target features specified at inference time.

### 2.3. Neural Encoders

In addition to selecting a minimum subset of features, one can summarize a dataset by training a neural network to extract key information.

Let $f : \mathcal{Z} \to \mathcal{T}$ be a neural encoder producing a representation $T = f(Z)$. Normally, the dimension of $T$ is significantly smaller than that of $Z$.

Earlier works, such as VAEs (Kingma & Welling, 2013) learn an encoding function $f$ through maximizing reconstruction fidelity and imposing regularization. In recent years, contrastive methods, such as SimCLR (Chen et al., 2020), define the learning criterion based on representation similarity. Given an anchor sample $Z$, and its "two views", $Z_1$ and $Z_2$, the loss

$$\mathcal{L}_{\text{SimCLR}} = -\mathbb{E}\left[ \log \frac{\exp(\langle f(Z_1), f(Z_2)\rangle/\tau)}{\sum_{Z' \neq Z_1} \exp(\langle f(Z_1), f(Z')\rangle/\tau)} \right],$$

measures similarity between representations under a temperature parameter $\tau > 0$, enforcing invariance across views of the same sample and discarding information not shared across them. Here, $Z'$ are views of samples in the same batch. More recently, masked autoencoders (MAEs) (He et al., 2022) learn representations by randomly masking part of the input and reconstructing the missing components. Given a random mask $\mathbf{m}$, let $(Z_{\mathbf{m}}, Z_{-\mathbf{m}})$ denote the observed and masked parts of $Z$. With encoder $f$ and decoder $g$, the objective is

$$\mathcal{L}_{\text{MAE}} = \mathbb{E}\left[\|Z_{-\mathbf{m}} - g(f(Z_{\mathbf{m}}))\|^2\right]. \tag{1}$$

In this paper, we propose a flow-inspired encoding method that ensures conditional sufficiency. Our method enforces an equality between two conditional distributions through the zero-flow criterion introduced in the next section.

## 3. Zero-flow Criterion and Conditional Independence

First, let us consider the general problem of learning sufficient information from a redundant feature set $Y$ to predict a target multi-dimensional variable $X$. Suppose $f$ is an "encoder", and $f(Y)$ contains the summary information of $Y$. The *sufficiency* of $f(Y)$ is certified using the following conditional independence

$$X \perp\!\!\!\perp Y | f(Y) \tag{2}$$

or equivalently,

$$p_{X|Y} = p_{X|f(Y)}. \tag{3}$$

The independence criterion above underpins the theoretical foundation of sufficient dimensionality reduction methods (Globerson & Tishby, 2003; Suzuki & Sugiyama, 2010; Chen et al., 2024a). These methods enforce conditions (2) or (3) by either solving a multivariate regression problem or maximizing a dependency criterion between $X$ and $f(Y)$.

In this section, we show that the independence (2) and the equality (3) are equivalent to a criterion, which we call the "zero-flow condition".

### 3.1. Zero-flow

Let $\mathbf{v}_t, t \in [0, 1]$ be a velocity field that transports a source distribution $p_X$ to a target distribution $p_{X'}$, meaning the following ODE

$$dX(t) = \mathbf{v}_t(X(t))dt \tag{4}$$

with the initial condition $X(0) = X \sim p_X$, has a solution $X(1) \sim p_{X'}$. (4) is the key to many generative models. There exist infinitely many $\mathbf{v}_t$ that transport from $p_X$ to

$p_{X'}$; much effort has been devoted to designing efficient flows (Klein et al., 2023; Xie et al., 2024; Geng et al., 2025).

Rectified flow learns $\mathbf{v}_t$ through the following objective function

$$\mathbf{v}_t := \arg\min_{\mathbf{u}_t} \int_0^1 \mathbb{E}\left[\|X' - X - \mathbf{u}_t(X_t)\|^2\right] \mathrm{d}t, \quad (5)$$

where $X_t = tX + (1-t)X'$. The population optimal to the above objective is $\mathbf{v}_t(\mathbf{z}) = \mathbb{E}[X' - X | X_t = \mathbf{z}]$. One curious question is that, if $p_X = p_{X'}$, is the velocity field zero everywhere? As we have seen in Figure 1, this is generally not the case. However, there is indeed a special property of the velocity field if $p_X = p_{X'}$.

**Theorem 3.1** (Zero-flow Condition). *If $X$ is independent of $X'$, $\mathbf{v}_{t=0.5}(\mathbf{z}) = \mathbf{0}, \forall \mathbf{z}$ if and only if $p_X = p_{X'}$.*

The proof can be found in Section B.1. This condition certifies the *perfect alignment* of $p_X$ and $p_{X'}$ and is central to our encoder methodology. In fact, Theorem 3.1 is a special case of the following antisymmetry.

**Theorem 3.2.** *(Antisymmetry) If $X$ is independent of $X'$, $\mathbf{v}_t(\mathbf{z}) = -\mathbf{v}_{1-t}(\mathbf{z}), \forall \mathbf{z}, \forall t \in [0,1]$, if and only if $p_X = p_{X'}$.*

This result is proven in Section C. In this paper, we only use Theorem 3.1 to develop the encoder. The application of Theorem 3.2 would be an interesting future work.

Recall that enforcing the conditional independence means aligning the *conditional distributions* in (3). Although Theorem 3.1 can test the equality of distributions, it cannot be directly used for testing the equality of two conditional distributions, as the velocity field is only trained on marginal distributions $p_X$.

In the next section, we propose a modified rectified flow whose optimal velocity field defines an ODE-based transport map between conditional distributions $p_{X|Y}$ and $p_{X|f(Y)}$, and prove that it also has a zero-flow condition.

### 3.2. Zero-flow Condition and Sufficient Information

Consider a $\mathbf{v}_t$ obtained from the following objective function:

$$\mathbf{v}_t := \arg\min_{\mathbf{u}_t} \int_0^1 \mathbb{E}\left[\|X' - X - \mathbf{u}_t(X_t, f(Y'), Y)\|^2\right] \mathrm{d}t, \quad (6)$$

where $X_t$ is the same as in (5) and $(X', Y')$ is an independent copy of $(X, Y)$.

Note that the above objective differs from existing conditional flows for sampling from a conditional distribution. The data do **not** come from different distributions (i.e., noise

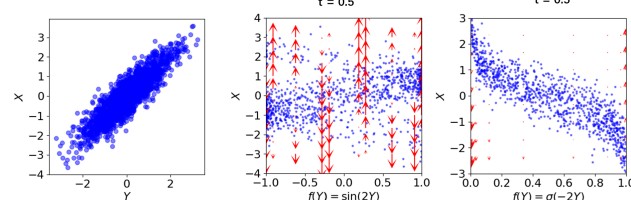

*Figure 2.* **Left**: $X = 0.5Y + \mathcal{N}(0,1)$. **Right**: $f(Y) = \sigma(-2Y)$ is a sufficient statistic for predicting $X$, thus $\mathbf{v}_{t=0.5} = 0$ almost everywhere as predicted by Theorem 3.4. **Center**: $f(Y) = \sin(2Y)$ is *not* a sufficient statistic for predicting $X$, thus $\mathbf{v}_{t=0.5} \neq 0$, as predicted by Theorem 3.4. The vector fields are trained using a two-layer ReLU network with an empirical version of (6).

and target). Both samples $(X', Y')$ and $(X, Y)$ are from *the same distribution*.

We can see that the optimizer of (6) has a closed form

$$\mathbf{v}_t(X_t; \eta, \xi) := \mathbb{E}[X' - X \mid X_t, \ f(Y') = \eta, \ Y = \xi,]. \quad (7)$$

We prove that the optimal velocity field (7) can indeed transport the distribution from $p_{X|Y}$ to $p_{X|f(Y)}$.

**Theorem 3.3** (Conditional Transport). *Let $X(0) = X \sim p_{X|Y}$, and let $X(t)$ solve the ODE*

$$\mathrm{d}X(t) = \mathbf{v}_t(X(t); f(Y), Y) \, \mathrm{d}t, \qquad t \in [0,1],$$

*where $\mathbf{v}_t$ is defined in (7), then $X(1) \sim P_{X|f(Y)}$.*

The proof can be found in Section B.3. The zero-flow condition for $\mathbf{v}_t(\cdot; \eta, \xi)$ is stated as follows.

**Theorem 3.4** (Conditional Zero-flow Condition). *For all pairs $(\xi, \eta)$ such that $f(\xi) = \eta$, and for all $\mathbf{z}$, the velocity field $\mathbf{v}_{t=0.5}(\mathbf{z}; \eta, \xi) = \mathbf{0}$ if and only if $p_{X|Y} = p_{X|f(Y)}$.*

The proof can be found in Section B.4.

Crucially, Theorem 3.4 implies that $\mathbf{v}_{t=0.5}(\cdot; f(Y), Y) = \mathbf{0}$ is synonymous with the equality (3) which defines the sufficiency of $f$ for predicting $X$. Therefore, it establishes the velocity field $\mathbf{v}_{t=0.5}$ as a rigorous mathematical tool for testing the sufficiency of $f$. We show a visualization in Figure 2.

**The idea behind *the zero-flow encoder*** is to learn a sufficient representation $f(Y)$ by minimizing the norm of velocity field $\mathbf{v}_{t=0.5}$ while restricting $f$ in some family (such as functions with sparse outputs).

In the following section, we turn this intuition into a concrete, tractable loss for enforcing the equality in (3).

## 3.3. Enforcing Conditional Independence with Zero-flow Loss

We propose the following least-squares minimization problem as the zero-flow representation learning criterion:

$$\min_f \mathbb{E}\|\mathbf{v}_{t=0.5}(X_t \, ; Y, f(Y))\|^2. \tag{8}$$

Note that directly imposing the zero-flow condition is computationally intractable as we need to enforce it over all $\mathbf{z}$. Here, we heuristically replace it with $X_t$, as it works well across all experiments. Combining (6) and (8), we obtain the following simulation-free least-squares objective:

$$\min_{\mathbf{u}, f \in \mathcal{F}} L(\mathbf{u}, f) = \underbrace{\int_0^1 \omega(t) \mathbb{E}\left[\|\mathbf{u}_t(X_t, f(Y), Y)\|^2\right] \mathrm{d}t}_{\text{zero-flow criterion}}$$
$$+ \underbrace{\int_0^1 \mathbb{E}\left[\|X' - X - \mathbf{u}_t(X_t, f(Y'), Y)\|^2\right] \mathrm{d}t}_{\text{rectified flow loss}}, \tag{9}$$

where $\omega(t) \geq 0$ is a time weighting function that peaks at $t = 0.5$ (e.g. a Laplace distribution centred at 0.5). It also balances the rectified loss and the zero-flow loss. $\mathcal{F}$ is the chosen family for our encoder. Some more concrete examples will be introduced in the following sections. The expectations in (9) can be approximated via samples. In practice, we approximate the IID samples of $(X', Y')$ by drawing bootstrap samples from $(X, Y)$ with replacement.

We refer to (9) as the *zero-flow loss* and the learned encoder $f$ as the *zero-flow encoder*. The zero-flow representation learning provides a general recipe for many sufficient information learning problems. Now, we introduce two applications of our zero-flow encoder.

# 4. Application: Learning Amortized Markov Blankets

As we discussed in Section 2.2, the Markov blanket is a succinct representation of a high-dimensional, complex graphical model. In this section, we propose a non-parametric estimation method for Markov blankets using our zero-flow encoder.

## 4.1. Learning Markov Blanket with Zero-flow Encoder

Suppose $Z = [Z_1, \ldots Z_d]$ is a $d$-dimensional undirected graphical model and $\mathbf{m} \in \{0, 1\}^d$ is a mask that partitions the random variables in $Z$ into two groups $[Z_\mathbf{m}, Z_{-\mathbf{m}}]$.

We now demonstrate the identification of a Markov blanket via a zero-flow encoder. We formulate the encoder loss by substituting $X = Z_\mathbf{m}, Y = Z_{-\mathbf{m}}$ in the objective (9). The

encoder is parameterized as $f_\mathbf{w}(Y) = Y \circ \sigma(\mathbf{w})$, where $\sigma$ is sigmoid function and $\mathbf{w} \in \mathbb{R}^d$ is the parameter. In this formulation, $\sigma(\mathbf{w})$ acts as a "gating function" that effectively performs feature selection on $Y$.

Theorem 3.4 ensures that the optimal subset $f_\mathbf{w}(Z_{-\mathbf{m}})$ selected by minimizing this loss satisfies the conditional independence condition

$$Z_\mathbf{m} \perp\!\!\!\perp Z_{-\mathbf{m}} \mid f_\mathbf{w}(Z_{-\mathbf{m}}).$$

To identify the Markov blanket, we need to ensure the selected subset is minimum, which can be achieved by regularizing the sparsity of gates $\sigma(\mathbf{w})$ in addition to the zero-flow loss:

$$\min_{\mathbf{u}, \mathbf{w} \in \mathbb{R}^d} L(\mathbf{u}, f_\mathbf{w}, \mathbf{m}) + \lambda \sum_{i=1}^d \sigma_i(\mathbf{w}).$$

Our method does not rely on any parametric assumptions about the underlying distribution of $Z$, and thus does not need to handle the troublesome normalisation term typically associated with the parametric graphical model estimation (Wainwright et al., 2008).

## 4.2. Amortized Markov Blanket Encoder

In existing literature, subset selection is typically performed offline by learning $\mathcal{M}(Z_\mathbf{m})$ for a fixed partition $\mathbf{m}$. However, the partition $\mathbf{m}$ may change at inference time. For instance, rather than analyzing standard rectangular patches, clinicians may want to manually select a specific region of interest within a medical image to inspect its dependencies. Given the computational constraints of large datasets, training a selector on demand is infeasible. In high-dimensional settings, the number of possible partitions is combinatorial, making it impossible to pre-train a model for every potential configuration of $\mathbf{m}$.

Moreover, it makes sense to share information when learning $\mathcal{M}(Z_\mathbf{m})$ for different $\mathbf{m}$. Suppose $Z$ is a time series. $Z_\mathbf{m}$ and $Z_{\mathbf{m}'}$ are two adjacent time windows. The Markov blanket of $Z_\mathbf{m}$ may have a similar structure to that of $Z_{\mathbf{m}'}$.

To this end, we can further generalize the encoder model $f$ to incorporate the information from the mask $\mathbf{m}$, thereby creating an amortized Markov blanket encoder. Specifically, we parametrize an *amortized encoder* $f_\beta(\mathbf{y}, \mathbf{m}) = \mathbf{y} \circ \sigma_\beta(\mathbf{m})$, where $\sigma_\beta(\mathbf{m})$ is a gating network. This design allows us to incorporate inductive bias based on the type of data we are dealing with. For example, if $Z$ is a time series, we can employ an LSTM as the gating network to ensure the temporal smoothness. We slightly modify the loss to be

$$\min_{\mathbf{u}, \beta} \mathbb{E}_M\left[ L(\mathbf{u}, f_\beta, M) + \lambda \sum_{i=1}^d [\sigma_\beta(M)]_i \right], \tag{10}$$

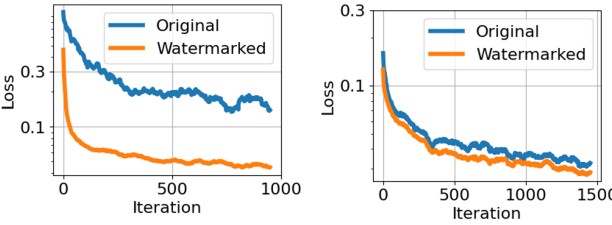

*Figure 3.* **Left**: Saturated SimCLR loss. On the watermarked CIFAR10 dataset, the loss quickly collapses to small values due to shortcuts. **Right**: The loss of the proposed method. The loss remains steady before and after the watermark is introduced.

where $M$ is a random draw of masks, which can be simulated from a Bernoulli distribution or a strategy that is more fitting to the downstream applications. In our experiments, we construct the target set $Z_M$ by simulating potential variables of interest. Specifically, given a time series $Z = [Z_1, \ldots, Z_T]$, we assume the user seeks the Markov blanket for a sequence of $k$ consecutive variables. We therefore define $Z_M$ as a randomly selected time window $Z_{[i:i+k]} = [Z_i, Z_{i+1}, \ldots, Z_{i+k-1}]$, where the starting index $i$ is drawn uniformly from $\{1, \ldots, T-k+1\}$.

Note that it is not straightforward to perform a similar amortized learning when training a parametric model for $p_Z$ using Maximum Likelihood Estimation or Score Matching. For each partition $Z = [Z_{\mathbf{m}}, Z_{-\mathbf{m}}]$, we need to specify and train a different model $p_{Z,\mathbf{m}}$. The number of models to train can be prohibitively large, unless we adopt some simplifications (such as a pairwise graphical model). Our method does not require such an assumption.

# 5. Application: Self-supervised Learning

Self-supervised learning (Liu et al., 2021; Xu et al., 2024) learns a latent representation of the dataset for downstream tasks. Unlike classic unsupervised methods, self-supervised learning focuses on learning semantic information from the dataset without relying on any handcrafted probabilistic models. Instead, it relies on assumptions about how different transformations (such as corruptions and image rotations) preserve global semantic information.

We rephrase a popular assumption and explain how the zero-flow encoder can exploit it to learn a representation.

## 5.1. Multi-view Assumption and Contrastive Learning

Let $Z_1$ and $Z_2$ be two augmented views of the same image, and $T$ be a latent, shared information. The multi-view assumption (Tian et al., 2020a) refers to the following data-generating scheme:

$$Z_2 \leftarrow T \rightarrow Z_1. \tag{11}$$

This assumption has been recognized as the foundation of the popular self-supervised learning algorithm SimCLR (Chen et al., 2020), which learns an encoding function $f$ that maps $Z_1$ and $Z_2$ back to their common cause $T$. This is achieved by maximizing the Mutual Information $\mathrm{MI}(f(Z_1), f(Z_2))$ or other affinity metrics (such as cosine similarity). Due to the data processing inequality and the conditional independence implied by (11), we can see that

$$\mathrm{MI}(f(Z_1), f(Z_2)) \leq \mathrm{MI}(Z_1, Z_2) \leq \mathrm{MI}(Z_1, T). \tag{12}$$

This means that SimCLR is maximizing a lower bound on the shared information between the view and the global information $T$ under the conditional independence encoded in the graphical model (11). See e.g., Zhang et al. (2023); Chen et al. (2024b) for more discussions.

## 5.2. Shortcut Problem

It is well-known that SimCLR suffers from the "shortcut problem" due to the *greediness of mutual information maximization* (Tian et al., 2020b; Robinson et al., 2021). The contrastive loss can be easily saturated by learning superficial shared features between $Z_1$ and $Z_2$ (a.k.a., shortcuts), causing the algorithm to fail to learn more meaningful semantic features (e.g., "dogs" or "horses"). We demonstrate such a case in Figure 3, where we train SimCLR on CIFAR-10. For each image, we add a instance-specific watermark patch in the top-left corner, which acts as a shortcut feature shared between views. It can be seen that the SimCLR loss quickly drops to zero once such a shortcut is learned. SimCLR is prone to shortcuts because its contrastive objective is essentially an instance-wise classification loss. Once an easy cue is found to separate the positive from the negative pairs, the loss saturates, leaving little incentive to learn additional semantic features. In contrast, without watermark patches, the loss decreases much more slowly, indicating that the model learn non-trivial, high-level patterns.

Rather than relying on mutual information maximization, we aim to enforce the conditional independence in (11) more directly using the zero-flow criterion. Similar to Section 4, let us reformulate the loss (9) by substituting $X$ with $Z_1$ and $Y$ with $Z_2$, assuming that the zero-flow criterion can be minimized to zero, then Theorem 3.4 ensures the conditional independence

$$Z_1 \perp\!\!\!\perp Z_2 \mid f(Z_2). \tag{13}$$

Assume that the shared information $T$ is recoverable from $Z_2$, the conditional independence of (13) is exactly the one that is encoded in (11). Note that, to avoid a trivial solution $f(Z_2) = Z_2$, we need to implement some information bottleneck. In our experiments, $f$ maps $Z_2$ to an $L$-dimensional latent space, and $L \ll d$. More details about this encoder architecture can be found in Section F.2.1.

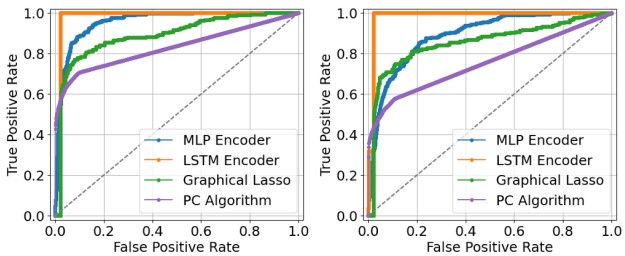

*Figure 4.* The comparison of ROC curves in terms of graphical model structure recovery on truncated-Gaussian (left) and non-paranormal (right) graphical models.

*Table 1.* AUC Performance in terms of graphical model structure recovery on Gaussian, nonparanormal, and truncated-Gaussian graphical models. Results are averaged over 10 random trials.

| Dataset | MLP (Ours) | LSTM (Ours) | PC-Fisher's Z | GLasso |
|---|---|---|---|---|
| Gaussian | $0.97 \pm 0.00$ | $\mathbf{0.98 \pm 0.00}$ | $0.85 \pm 0.01$ | $0.94 \pm 0.01$ |
| Nonparanormal | $0.79 \pm 0.02$ | $\mathbf{0.97 \pm 0.01}$ | $0.75 \pm 0.01$ | $0.78 \pm 0.02$ |
| Truncated | $0.95 \pm 0.01$ | $\mathbf{0.98 \pm 0.00}$ | $0.83 \pm 0.01$ | $0.88 \pm 0.02$ |

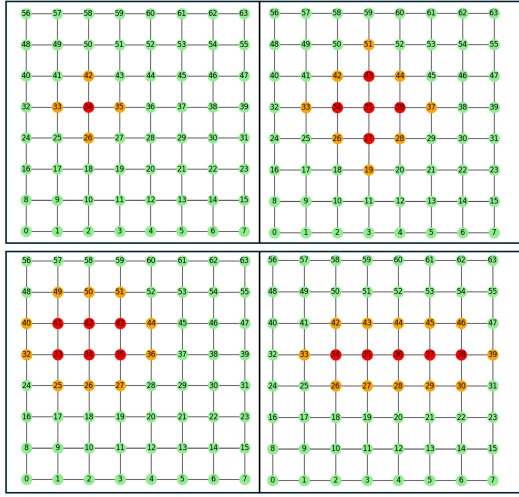

*Figure 5.* The identified Markov blankets (yellow) of target variables (red). The true graphical model (lattice) is visualized for convenience. The true Markov blanket is the set of neighbouring nodes of the red in the lattice. **Top**: Target variables observed during training. **Below**: Targets were unseen during training.

In Section 6.2, we show that, although derived from the same assumption, our zero-flow encoder does not suffer from the shortcut problem.

## 6. Experiments

### 6.1. Learning Markov Blankets

In this suite of experiments, we evaluate the ability of the zero-flow encoder to recover Markov blankets. Detailed experimental settings are provided in Appendix D, with additional results in Appendix E.

#### 6.1.1. Learning Structure of Graphical Model

We first test our encoder using the task of recovering the structure of sparse *non-Gaussian* graphical models.

We generate 2048 samples of $\tilde{Z}$ from a normal distribution $\mathcal{N}(\mathbf{0}, \Theta^{-1})$, where $\Theta \in \mathbb{R}^{64 \times 64}$ is a sparse precision matrix. $\tilde{Z}$ is a 64-dimensional undirected graphical model whose structure is determined by the non-zero elements in $\Theta$. We then apply non-linear transforms $Z = T(\tilde{Z})$ to create non-Gaussian graphical models: Nonparanormal and Truncated Gaussian. Each transform preserves the conditional independence structure encoded in $\Theta$. The task of graphical model learning is to reconstruct the sparsity pattern of $\Theta$, given samples of $Z$. For the first set of experiments, we set the graphical model to a 3rd-order Markov chain.

To recover the structure using zero-flow encoder, we first train a Markov blanket encoder $f$ using (10). Then, we create a gate matrix $G$, where $G_{i,j} = [\sigma_{\hat{\beta}}(\mathbf{m}^{(i)})]_j$ and $\mathbf{m}^{(i)}$ is an all-zero vector with the $i$-th element set to one. $\sigma_{\hat{\beta}}(\mathbf{m}^{(i)})$ encodes the Markov blanket of $Z_i$ and $G_{i,j} \neq 0$ indicate $Z_j \in \mathcal{M}(Z_i)$. If a zero-flow encoder can recover the correct graphical model structure, we should observe that the gate matrix $G$ has the same non-zero pattern as $\Theta$.

We test two different architecture for the gating network $\sigma_{\hat{\beta}}$: a two-layer MLP and an LSTM. The latter architecture introduces a sequential inductive bias, assuming that the Markov blanket of a variable $Z_i$ is likely to comprise locally adjacent neighbors, such as $Z_{i-1}$ and $Z_{i+1}$.

We compare our method with Graphical Lasso (Banerjee et al., 2008), which learns the precision matrix $\Theta$ by $\ell_1$-regularized maximum likelihood, and with the PC algorithm from `causal-learn` (Zheng et al., 2024), a constraint-based method that infers graph structure via Fisher's $Z$ conditional independence tests. Note that we attempted using the kernel-based non-parametric test criterion `kci` provided by the package. However, it is estimated to run on our dataset for 24 hours for a single seed on CPU. In comparison, our method is also non-parametric, but it trains an amortized MLP encoder in 30 seconds on the same CPU.

The ROC curve for recovering the graphical model structure is plotted in Figure 4. We repeat the experiments 10 times with different random seeds, and show the mean AUC values in Table 1. It can be seen that zero-flow encoder outperforms graphical lasso and the PC algorithm in both non-Gaussian settings. Notably, the LSTM-based encoder achieves near-perfect AUC scores by effectively leveraging the appropriate inductive bias. Our further experiments show that CNN based encoder also works well on lattice networks, even when the dimensions are large. See Section E for details.

Finally, we evaluate whether the learned encoder can amor-

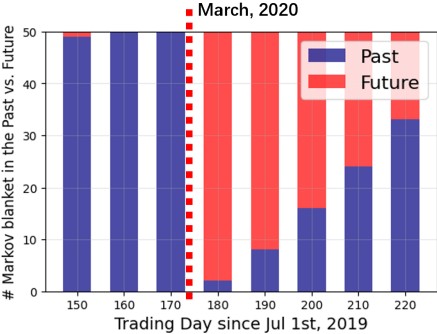

*Figure 6.* The sudden change of Markov blanket patterns, before and after March 2020. Before, Markov blankets consisted of past trading days. Immediately after, Markov blankets predominantly consisted of future trading days.

tize Markov blanket discovery across different target sets on a lattice graphical model. The encoder is trained using randomly sampled target sets consisting of a node and, with probability $50\%$, each of its four lattice neighbours. We identify $Z_j \in \mathcal{M}(Z_{\mathbf{m}})$ whenever $[\sigma_{\hat{\beta}}(\mathbf{m})]_j > 0.1$. At inference time, we evaluate both in-sample targets observed during training and out-of-sample targets that were never used for training. As shown in Figure 5, the zero-flow encoder correctly recovers the Markov blankets in both cases, demonstrating that the learned amortized encoder generalizes to unseen target variables.

### 6.1.2. MARKOV BLANKETS ON S&P500 IN 2019-2020

The Markov blanket is often used to interpret data. Here, we apply our method to S&P 500 data to detect changes in time series. We obtain S&P 500 stock prices for the period from *July 1st, 2019*, to *June 30th, 2020*. We treat each stock as a sample and each day as a feature. Thus, the dataset consists of 500 samples and 252 features.

We run (10) to learn the amortized Markov blanket of $Z_{\mathbf{m}}$ for any five consecutive trading days during this period. Since each trading day is a feature, the Markov blanket consists of trading days that are most predictive of these five-day windows.

Under the assumption that the market is stable, the past should be the most predictive of its present. However, this assumption can be violated when the market is disrupted by major ongoing events, making the past less predictive. This can be detected from a shifting pattern in Markov blankets. As the market is being disrupted, the number of past days in the Markov blanket reduces significantly, and future days will dominate. In fact, measuring the predictiveness of the past samples (using autoregressive models) is a classic change-point detection strategy (Takeuchi & Yamanishi, 2006) and identifying changes in graphical models to detect changes in the dataset has also been well-established (Liu

et al., 2017; Kim et al., 2021).

In Figure 6, we plot the percentage of past and future days in the Markov blankets learned by zero-flow encoder. Here, we select the Markov blanket as the features corresponding to the top-50 largest gating values, $\sigma_{\hat{\beta}}(\mathbf{m})$. We can see the number of past days in the Markov blanket drops sharply after March, 2020, signalling a suddenly disrupted market following the COVID-19 pandemic outbreak in the US.

### 6.2. No Shortcuts!

In this experiment, we evaluate the zero-flow encoder on a range of image datasets and compare it with SimCLR. Our goal is not to outperform SimCLR in standard state-of-the-art settings, but rather to demonstrate that the zero-flow encoder is less vulnerable to the shortcut problem, despite relying on the same multi-view assumption discussed in Section 5.1.

For a fair comparison, we use *the same encoder architecture* (3 CNN layers and an MLP projection head) for both methods. Details of these experiments are in Section F.

First, we evaluate the performance on Fashion-MNIST (Xiao et al., 2017) and CIFAR-10 (Krizhevsky et al., 2009) datasets with artificial shortcuts. We construct artificial shortcuts by adding meaningless instance-specific watermark information. For Fashion-MNIST, we assign each image a random color. For CIFAR-10, we add a small random $10 \times 10$ patch to the top-left corner of each image. These shortcuts provide an easy cue for contrastive learning, allowing SimCLR to distinguish positive pairs from negatives without learning semantic image features.

As shown in Figure 3, SimCLR suffers from this shortcut issue: once the watermark is introduced, its loss rapidly saturates. In contrast, the zero-flow loss remains stable before and after introducing the watermark. Figure 8 shows that, on Fashion-MNIST, the zero-flow encoder continues to encode clothing shapes even when color shortcuts are present. This suggests that the shortcuts do not dominate the learned representation.

Now we move on to larger scale datasets (STL-10 (Coates et al., 2011), TinyImageNet (Le & Yang, 2015), ImageNet-1K (Russakovsky et al., 2015)) and evaluate the learned representations using downstream applications. Similar to CIFAR10 dataset, we preprocess the datasets by adding a random patch to the top left corner of the images, covering 1/9 area of the image, as the shortcut. We consider two applications: linear probing and image reconstruction.

We report linear probing accuracy on the test set in Table 2, together with reconstruction MSE. We also include the change of performance comparing to the same task on a clean, original dataset. We include a Masked Autoencoder

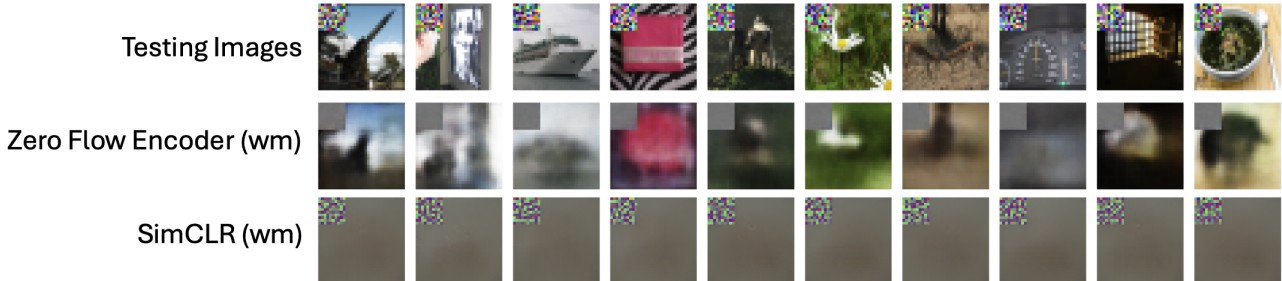

*Figure 7.* Reconstruction of watermarked ImageNet-1K images. The top row shows the original testing images with random watermark patches in the top-left corners. The second and the third row show images reconstructed from representations extracted using our encoder and SimCLR, respectively.

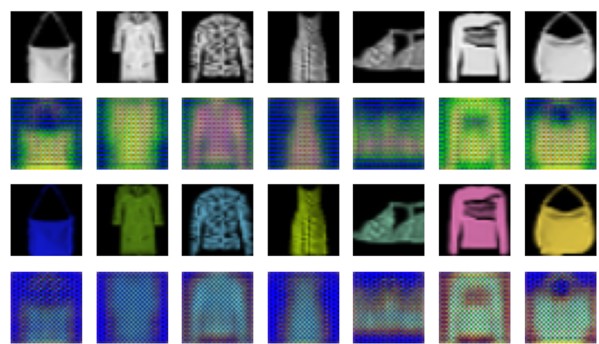

*Figure 8.* Fashion MNIST with and without random colors. The learned encodings (row 2 and 4) vs. the original images (row 1 and 3). It can be seen that zero-flow encoding continues to learn semantic information (e.g., the shapes of clothes) despite shortcut information (color).

*Table 2.* Comparison of linear probing accuracy and reconstruction MSE across **watermarked datasets**. Values in parentheses indicate change in performance compared with the clean, non-watermarked dataset. The best performance is highlighted.

| Dataset | Method | Accuracy | Recon. MSE |
|---|---|---|---|
| STL-10 | Ours | 52.21% (+2.41) | 0.0256 (−0.0003) |
| STL-10 | SimCLR | 10.00% (−60.23) | 0.0674 (+0.0210) |
| STL-10 | MAE | **56.14%** (+0.65) | **0.0245** (−0.0001) |
| TinyImageNet | Ours | 16.60% (+0.50) | **0.0290** (+0.0005) |
| TinyImageNet | SimCLR | 0.50% (−30.63) | 0.0758 (+0.0267) |
| TinyImageNet | MAE | **18.75%** (−0.24) | 0.0305 (−0.0007) |
| ImageNet-1K | Ours | 7.01% (+0.82) | **0.0161** (−0.0003) |
| ImageNet-1K | SimCLR | 0.10% (−15.18) | 0.0600 (+0.0361) |
| ImageNet-1K | MAE | **8.1%** (−0.2) | 0.0177 (−0.0025) |

(MAE) with a Vision Transformer backbone (Dosovitskiy et al., 2021; He et al., 2022) as an additional strong baseline. We can see that the performance degradation of SimCLR is substantial, indicating that the learned representation is dominated by the shortcut. In contrast, our method and MAE are more robust, showing much smaller changes in accuracies and reconstruction errors.

Figure 7 provides visual reconstruction examples on unseen watermarked images. Both encoders capture the watermark in the top-left corner, indicating that the shortcut information is encoded. However, SimCLR only reconstructs the watermark and fails to preserve meaningful visual structures of the original image. In contrast, our representation retains richer semantic information beyond the shortcut, consistent with the observation in Figure 8.

## 7. Limitations

In Section 3.3, we used heuristics to derive zero-flow loss, enforcing $\mathbf{v}_t = 0$ only at $X_t$. Although it works well in experiments, a more rigorous justification is needed. Compared with other encoding techniques, such as VAE, Sim-CLR, and MAE, the zero-flow encoder requires a velocity field model $\mathbf{v}_t$ in addition to the encoder, which incurs additional modeling and computational costs. Rectified flow is designed to transport continuous-valued distributions, thus zero-flow encoder is not directly applicable to discrete variables.

## 8. Conclusion and Future Works

In this paper, we introduced zero-flow, a novel criterion inspired by inspecting velocity fields of rectified flow models. We theoretically established that this was both necessary and sufficient for conditional independence. By transforming this theoretical insight into a tractable zero-flow-loss formulation, we enabled the extraction of sufficient information in a non-parametric fashion. We demonstrated the efficacy of this framework in two applications—amortized Markov blanket learning and robust self-supervised learning—where it consistently achieved competitive performance and validated our theoretical claims.

One promising future direction is to first map the target variable $X$ to a latent space then perform zero-flow encoding in this latent space. This extension allows us to handle high-dimensional or discrete target data.

## Impact Statement

This paper presents work whose goal is to advance the field of Machine Learning. There are many potential societal consequences of our work, none which we feel must be specifically highlighted here.

## Acknowledgements

We thank Sam Power for helpful discussions. TS was partially supported by JSPS KAKENHI (24K02905) and JST CREST (PMJCR2015). This research is supported by the National Research Foundation, Singapore and the Ministry of Digital Development and Information under the AI Visiting Professorship Programme (award number AIVP-2024-004). Any opinions, findings and conclusions or recommendations expressed in this material are those of the author(s) and do not reflect the views of National Research Foundation, Singapore and the Ministry of Digital Development and Information. The authors thank three anonymous reviewers for their insightful feedback.

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

## A. An Illustrative Example of the Zero-flow Phenomenon

Consider a distribution $p_X$. From this distribution, we construct two independent batches of particles, $X = \{x_i\}_{i=1}^m$ and $X' = \{x_i'\}_{i=1}^m$, both sampled i.i.d. from $p_X$. These batches are paired via **independent coupling** (Figure 9(a)). A velocity field model $\mathbf{v}_t$ is trained using rectified flow objective (5).

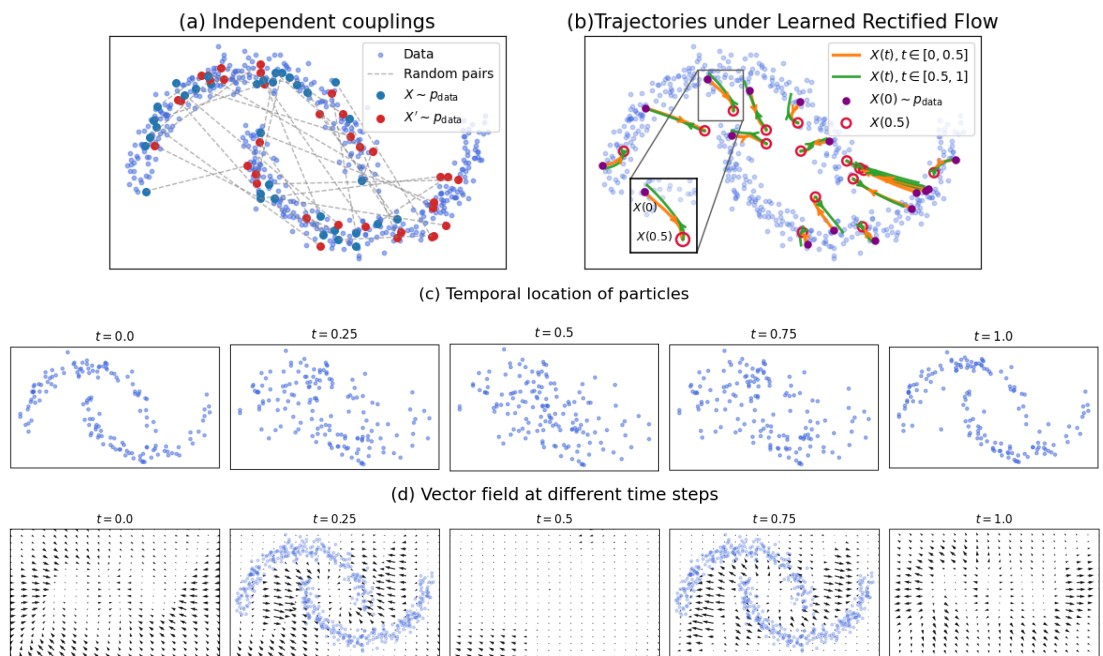

*Figure 9.* **Demonstration of the zero-flow property.** **(a)** Independent coupling between $X$ and $X'$. **(b)** Trajectories $X(t)$. **(c)** Temporal snapshots of particles. **(d)** Learned velocity fields $\mathbf{v}_t$ at various $t$, highlighting the vanishing vector field at $t = 0.5$.

As shown by the orange trajectories in Figure 9(b), despite the fact that the endpoint marginals satisfy $p_X = p_{X'}$, the flow is *not* zero. This non-static behaviour arises from the stochasticity introduced by independent coupling: individual particles are transported toward an intermediate position before returning to the support of the original position. Figure 9(c) and (d) visualizes this process.

Moreover, this convergence-divergence pattern reflects a fundamental **antisymmetry** in the induced flow. The trained velocity field must satisfy $\mathbf{v}_t(\mathbf{z}) = -\mathbf{v}_{1-t}(\mathbf{z})$ for all $t \in [0, 1]$. This property follows from the symmetry of independent coupling (See Theorem 3.2 for details). As a direct consequence, the expected velocity at the temporal midpoint must vanish, $\mathbf{v}_{0.5}(\mathbf{z}) = 0$, which we refer to as the *zero-flow* condition. Figure 9(d) provides a direct visualization of this phenomenon, showing that the learned velocity fields are indeed antisymmetric and collapse to near zero magnitude at $t = 0.5$.

## B. Proofs and Additional Theorems

### B.1. Proof of Theorem 3.1

*Proof.* First, we prove that if $p_X = p_{X'}$ then $\mathbf{v}_{t=0.5}(\mathbf{z}) = \mathbf{0}$. We can see that

$$\mathbf{v}_{t=0.5}(\mathbf{z}) = \mathbb{E}[X' - X | \frac{X'}{2} + \frac{X}{2} = \mathbf{z}]$$
$$= \mathbb{E}[X' - X | X' + X = 2\mathbf{z}]. \tag{14}$$

Since $X'$ and $X$ are independent and identically distributed, due to symmetry, we have

$$\mathbb{E}[X' | X' + X = 2\mathbf{z}] = \mathbb{E}[X | X + X' = 2\mathbf{z}]$$

for all $\mathbf{z}$. Hence, (14) is zero because of the linearity of expectations. The opposite direction holds due to Lemma B.1. $\square$

## B.2. Characterization of Identical Distributions via Midpoint Velocity

**Lemma B.1.** *Let $X_0 \sim p_{X_0}$ and $X_1 \sim p_{X_1}$ be two independent random variables in $\mathbb{R}^d$ with non-vanishing characteristic functions. Let $\mathbf{v}(z,t)$ be the optimal Rectified Flow velocity field defined by the conditional expectation:*

$$\mathbf{v}(z,t) = \mathbb{E}[X_1 - X_0 \mid (1-t)X_0 + tX_1 = z]. \tag{15}$$

*If $\mathbf{v}(z, 0.5) = 0$ everywhere, then $p_{X_0} = p_{X_1}$.*

*Proof.* $\mathbf{v}(z, 0.5) \equiv 0$ implies that for all $z$:

$$\mathbb{E}\left[X_1 - X_0 \;\middle|\; \frac{X_0 + X_1}{2} = z\right] = 0. \tag{16}$$

This is equivalent to conditioning on $S = X_0 + X_1 = s$:

$$\mathbb{E}[X_1 - X_0 \mid S = s] = 0. \tag{17}$$

It implies for any measurable function $h$,

$$\mathbb{E}[(X_1 - X_0)h(S)] = 0. \tag{18}$$

Let $h(S) = e^{itS} = e^{it(X_1 + X_0)}$, and

$$\mathbb{E}[(X_1 - X_0)e^{it(X_1+X_0)}] = \mathbb{E}[X_1 e^{it(X_1+X_0)}] - \mathbb{E}[X_0 e^{it(X_1+X_0)}] = 0. \tag{19}$$

Moreover, due to the independence of $X_1$ and $X_0$

$$\mathbb{E}[X_1 e^{it(X_1+X_0)}] = \mathbb{E}[X_1 e^{it(X_1)}]\mathbb{E}[e^{it(X_0)}]. \tag{20}$$

Thus (19) can be written as

$$\mathbb{E}[X_1 e^{it(X_1)}]\mathbb{E}[e^{it(X_0)}] = \mathbb{E}[X_0 e^{it(X_0)}]\mathbb{E}[e^{it(X_1)}]. \tag{21}$$

Notice that $\phi_1(t) = \mathbb{E}[e^{it(X_1)}]$ is the characteristic function of $p_{X_1}$ and $\phi_0(t) = \mathbb{E}[e^{it(X_0)}]$ is the characteristic function of $p_{X_0}$. $\phi_1'(t) = i\mathbb{E}[X_1 e^{it(X_1)}]$ and $\phi_0'(t) = i\mathbb{E}[X_0 e^{it(X_0)}]$. Thus, (21) can be expressed as

$$\phi_1(t)\phi_0'(t)/i = \phi_0(t)\phi_1'(t)/i. \tag{22}$$

Rewrite (22) and integrate on both sides:

$$\frac{\phi_1'(t)}{\phi_1(t)} = \frac{\phi_0'(t)}{\phi_0(t)}, \implies \log \phi_1(t) = \log \phi_0(t) + C. \tag{23}$$

Since $\phi_0(0) = \phi_1(0) = 1 \implies C = 0$. It further implies $\phi_0(t) = \phi_1(t)$, i.e., $X_0$ and $X_1$ have the same characteristic function. The desired result follows.

$\square$

## B.3. Proof of Theorem 3.3

**Theorem B.2** (Restatement Theorem 3.3)**.** *Consider the conditional rectified-flow objective*

$$\mathbf{v}_t \in \arg\min_{\mathbf{u}_t} \int_0^1 \mathbb{E}\left[\left\|X' - X - \mathbf{u}_t(X_t, Y, f(Y'))\right\|^2\right] dt. \tag{24}$$

*Then for each $t \in [0,1]$ an optimal velocity field admits the population form*

$$\mathbf{v}_t(x; y, z') := \mathbb{E}\left[X' - X \mid X_t = x,\ Y = y,\ f(Y') = z'\right]. \tag{25}$$

*Let $X(0) = X$ with $X \sim p_{X|Y=y}$, and let $X(t)$ solve the ODE*

$$dX(t) = \mathbf{v}_t(X(t); y, z)\, dt, \qquad t \in [0, 1], \tag{26}$$

*where $z = f(y)$.*

*Assume that the continuity equation with drift $x \mapsto \mathbf{v}_t(x; y, z)$ admits a unique solution. Then*

$$X(1) \sim p_{X|f(Y)=z}. \tag{27}$$

*Proof.* For each fixed $t \in [0, 1]$, the integrand in (24) is a squared loss in the value $u_t(X_t, Y, f(Y'))$. Hence the population minimizer is the conditional expectation of the target $X' - X$ given the input $(X_t, Y, f(Y'))$, which yields (25).

Set $z = f(y)$ and let

$$X_0 \sim p_{X|Y=y}, \quad X_1 \sim p_{X'|f(Y')=z}. \tag{28}$$

and assume $X_0$ and $X_1$ are independent. Define the linear interpolant

$$\widetilde{X}_t := (1-t)X_0 + tX_1, \qquad t \in [0, 1]. \tag{29}$$

Let $\rho_t(\,\cdot\,; y, z)$ denote the law of $\widetilde{X}_t$. For any smooth test function $\varphi : \mathbb{R}^d \to \mathbb{R}$ with compact support, differentiation of the expectation gives

$$\frac{d}{dt}\mathbb{E}\big[\varphi(\widetilde{X}_t)\big] = \mathbb{E}\big[\nabla\varphi(\widetilde{X}_t)^\top \frac{d}{dt}\widetilde{X}_t\big] = \mathbb{E}\big[\nabla\varphi(\widetilde{X}_t)^\top (X_1 - X_0)\big], \tag{30}$$

Conditioning on $\widetilde{X}_t$ and applying the tower property yields

$$\frac{d}{dt}\int \varphi(x)\,\rho_t(x; y, z)\, dx = \int \nabla\varphi(x)^\top \mathbf{b}_t(x)\,\rho_t(x; y, z)\, dx$$
$$= -\int \varphi(x)\nabla \cdot (\mathbf{b}_t(x)\,\rho_t(x; y, z))\, dx, \tag{31}$$

where $\mathbf{b}_t(x) := \mathbb{E}\Big[X_1 - X_0 \,\Big|\, \widetilde{X}_t = x\Big]$ and the second line follows from integration by parts since $\varphi$ has compact support. Moreover,

$$\frac{d}{dt}\int \varphi(x)\,\rho_t(x; y, z)\, dx = \int \varphi(x)\,\frac{d}{dt}\rho_t(x; y, z)\, dx. \tag{32}$$

Due to (31) and (32), $\rho_t(\,\cdot\,; y, z)$ and $\mathbf{b}_t$ satisfy the continuity equation with the initial condition

$$\rho_0(\,\cdot\,; y, z) = p_{X|Y=y}. \tag{33}$$

Notice that by construction, the conditional distribution of $(X, X')$ given $Y = y$, $f(Y') = z$ matches that of $(X_0, X_1)$. Thus,

$$\mathbf{b}_t(x) = \mathbb{E}[X' - X \mid X_t = x,\ Y = y,\ f(Y') = z] = \mathbf{v}_t(x; y, z),$$

which is the velocity field of $X(t)$.

Let $\nu_t(\,\cdot\,; y, z)$ denote the law of $X(t)$. By construction, $\nu_0(\,\cdot\,; y, z) = \rho_0(\,\cdot\,; y, z)$. The continuity equation and the uniqueness assumption imply that

$$\nu_t(\,\cdot\,; y, z) = \rho_t(\,\cdot\,; y, z) \qquad \text{for all } t \in [0, 1].$$

In particular,

$$X(1) \overset{d}{=} \widetilde{X}_1 \overset{d}{=} X_1 \sim p_{X'|f(Y')=z}. \tag{34}$$

Finally, since $(X', Y') \overset{d}{=} (X, Y)$, we conclude

$$X(1) \sim p_{X|f(Y)=z}, \tag{35}$$

which completes the proof. $\qquad\square$

### B.4. Proof of Theorem 3.4

**Theorem B.3** (Conditional Zero-flow Condition Restatement)**.** *For any pair $(\xi, \eta)$, if $f(\xi) = \eta$, the velocity field $\mathbf{v}_{t=0.5}(\cdot\,; \eta, \xi) = \mathbf{0}$ if and only if*

$$p_{X|Y} = p_{X|f(Y)}.$$

*Proof.* Let us we prove that if $p_{X|Y} = p_{X|f(Y)}$, $f(\xi) = \eta$, then $\mathbf{v}_{t=0.5}(\mathbf{z}; \eta, \xi) = \mathbf{0}, \forall \mathbf{z}$.

Let us define two conditional random variables

$$U := X|Y = \xi, \quad V := X'|f(Y') = \eta.$$

Due to the condition $p_{X|Y} = p_{X|f(Y)}$, we have $X|Y \overset{\text{d}}{=} X|f(Y)$, thus

$$U = X|\{Y = \xi\} = X|\{f(Y) = f(\xi)\} = X|\{f(Y) = \eta\},$$

where the last equality is due to the setting $f(\xi) = \eta$. Since $(X, Y) \overset{\text{d}}{=} (X', Y')$, we can see that $U \overset{\text{d}}{=} V$. Moreover, $(X, Y)$ and $(X', Y')$ are also independent, so $U, V$ are i.i.d.

Moreover, by definition

$$\mathbf{v}_{t=0.5}(\mathbf{z}; \eta, \xi) = \mathbb{E}[X' - X|X' + X = \mathbf{z}, f(Y') = \eta, Y = \xi] = \mathbb{E}[V - U|U + V = \mathbf{z}].$$

The proof for $\mathbb{E}[V - U|U + V = \mathbf{z}] = 0$ with i.i.d. $U, V$ follows from the sufficiency proof in Theorem 3.1.

We can also express the opposite direction using $U, V$. We want to prove that, for independent variables $U, V$, if

$$\mathbb{E}[V - U|V + U = \mathbf{z}] = 0, \forall \mathbf{z},$$

then $p_U = p_V$. This is proven in Lemma B.1.

□

## C. Anti-symmetric Characterization of Zero-Flow

In Section 3, we showed that the midpoint velocity field $\mathbf{v}_{t=0.5}$ provides both a necessary and sufficient criterion for distributional equality and conditional sufficiency. In this subsection, we further generalize this result by showing an anti-symmetry property of the rectified velocity field over time.

**Theorem C.1.** *Let $\mathbf{v}_t$ be the optimal velocity field learned by the Rectified Flow objective using independent coupling. $\mathbf{v}_t(\mathbf{z}) = -\mathbf{v}_{1-t}(\mathbf{z})$ for all $t \in [0, 1]$, if and only if $p_X = p_{X'}$.*

*Proof.* First we prove that if $p_X = p_{X'}$, then $\mathbf{v}_t = -\mathbf{v}_{1-t}, \forall t$. By definition,

$$\mathbf{v}_t(\mathbf{z}) = \mathbb{E}[X' - X|tX' + (1 - t)X = \mathbf{z}]$$
$$\mathbf{v}_{1-t}(\mathbf{z}) = \mathbb{E}[X' - X|(1 - t)X' + tX = \mathbf{z}]$$

If $p_X = p_{X'}$, $X$ and $X'$ are i.i.d. random variables, thus are exchangeable in the above formulas,

$$\begin{aligned}
\mathbb{E}[X' - X|tX' + (1 - t)X = \mathbf{z}] &= \mathbb{E}[X - X'|tX + (1 - t)X' = \mathbf{z}] \\
&= -\mathbb{E}[X' - X|(1 - t)X' + tX = \mathbf{z}] \\
&= -\mathbf{v}_{1-t}(\mathbf{z}).
\end{aligned} \tag{36}$$

The opposite is true is due to the fact that

$$\mathbf{v}_t(\mathbf{z}) = -\mathbf{v}_{1-t}(\mathbf{z}), \forall t \implies 2\mathbf{v}_{0.5}(\mathbf{z}) = 0.$$

As we stated in Theorem 3.1, this ensures that $p_X = p_{X'}$.

□

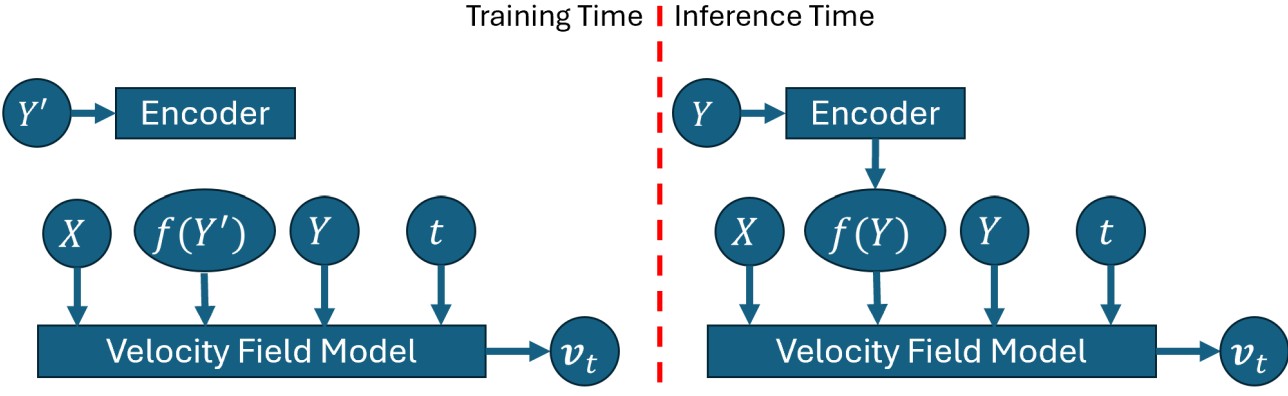

Figure 10. The neural network architecture for learning Markov blanket Section 6.1 during inference time.

## D. Implementation of Markov Blanket Learning Experiments in Section 6.1

This section describes the exact implementation details for the graphical model recovery experiments.

**Data generation.** We generate synthetic data from $d$-dimensional Gaussian graphical models with precision matrix $\Theta \in \mathbb{R}^{d \times d}$. Two classes of graph structures are considered.

*(i) Markov chain.* In the main setting, $\Theta$ corresponds to a $k$-th order Markov chain, i.e.,

$$\Theta_{ij} \neq 0 \quad \Longleftrightarrow \quad 0 < |i - j| \leq k,$$

with $k = 3$. The nonzero off-diagonal entries are assigned fixed weights $\{w_r\}_{r=1}^k = \{0.8, 0.4, 0.2\}$, and the diagonal is set to ensure strict diagonal dominance and positive definiteness.

*(ii) Lattice grid.* In the second setting, variables are arranged on a two-dimensional lattice of size $p \times p$ with $d = p^2$. Each node is connected to its four immediate spatial neighbors. All nonzero off-diagonal entries are assigned a constant weight, and the diagonal is again chosen to ensure positive definiteness.

In both cases, we define $\Sigma = (\Theta)^{-1}$ and generate a latent Gaussian vector

$$\tilde{Z} \sim \mathcal{N}(0, \Sigma).$$

From this latent model, we construct three data regimes used in the experiments:

*(a) Gaussian.* The standard setting directly uses $Z = \tilde{Z}$, yielding data drawn exactly from the Gaussian graphical model.

*(b) Non-paranormal.* To obtain non-Gaussian but conditionally independent data with the same copula structure, we apply a monotone marginal transformation

$$Z_i = \text{sign}(\tilde{Z}_i) \, |\tilde{Z}_i|^\gamma, \quad \gamma = 3,$$

followed by standardization to zero mean and unit variance for each coordinate. This preserves the underlying conditional independence graph while inducing heavy-tailed non-Gaussian marginals.

*(c) Truncated Gaussian.* To model distributional shift and selection bias, we generate data from the truncated distribution

$$Z = \tilde{Z} \mid \tilde{Z}_i > \tau \ \forall i,$$

with truncation threshold $\tau = -0.75$. This yields a non-Gaussian distribution whose support is restricted to a half-space, while retaining the same latent graphical structure.

All experiments use $d = 50$ for the Markov chain and $d = 64$ for the lattice grid, with training set size $n_{\text{train}} = 2{,}048$.

**Model architecture.** The model consists of an encoder network and a velocity field network. The schematic of this network is plotted in Figure 10. Note that there is a minor difference in the network inputs between training and inference. In the following text, we assume the inference time setting.

One of the encoder $f(Y; \mathbf{m})$ we consider is an MLP encoder, denoted as

$$\texttt{Encoder}(d, 128),$$

where $\sigma_\beta$ is a two-layer ReLU network with 128 hidden units.

The second type of encoder $f(Y; \mathbf{m})$ we consider is an LSTM encoder, where $\sigma_\beta$ is a bidirectional LSTM layer with 32 hidden units, and a linear projection layer mapping the output to $d$-dimensional. It is denoted as

$$\texttt{LSTMEncoder}(d, 32),$$

The third type of encoder is a CNN encoder, where $\sigma_\beta$ first reshapes $\mathbf{m}$ to a $\sqrt{d} \times \sqrt{d}$ image, then passes through three layers of CNN with 32 hidden channels, and RELU activation functions. The output from such a CNN network is flattened before returned.

$$\texttt{CNNEncoder}(d, 32).$$

The velocity field is an MLP with 256 hidden units, denoted as

$$\texttt{VectorField}(d, 256),$$

which takes as input the interpolation $X_t$, the encoder output $f(Y)$, $Y$, the mask $m$, and time $t$, and outputs a velocity prediction $\hat{v}_t \in \mathbb{R}^d$ during inference time.

**Rectified flow objective.** Given two independent samples $z_1, z_2 \sim \mathcal{N}(0, \Sigma^\star)$ and a mask $m$, we form

$$x = z_1 \odot m, \quad y = z_1 \odot (1 - m), \qquad x' = z_2 \odot m, \quad y' = z_2 \odot (1 - m).$$

We sample $t \in (0, 1)$ from a symmetric Beta distribution $t \sim \text{Beta}(\alpha, \alpha)$ with $\alpha = 4$ and define the interpolation

$$x_t = tx' + (1 - t)x.$$

The rectified flow loss is

$$\mathcal{L}_{\text{RF}} = \mathbb{E}\big[\|(x' - x - \hat{v}_t) \odot m\|_2^2\big].$$

**Zero-flow regularization.** To encourage vanishing velocity near the midpoint of the flow, we introduce a time-local penalty

$$\mathcal{L}_{\text{ZF}} = \mathbb{E}\big[\omega(t)\|\hat{v}_t \odot m\|_2^2\big],$$

where

$$\omega(t) = \exp(-|t - 0.5|/b)\cdot$$

with bandwidth $b = 5 \times 10^{-4}$.

**Gate sparsity.** We regularize the encoder gates using an $\ell_1$-style penalty

$$\mathcal{L}_{\ell_1} = \sum_{j=1}^{d} g_j,$$

weighted by $\lambda = 3 \times 10^{-9}$. This encourages sparse Markov blankets.

**Total training objective.** The full objective is

$$\mathcal{L} = \mathcal{L}_{\text{RF}} + \mathcal{L}_{\text{ZF}} + \lambda \mathcal{L}_{\ell_1}.$$

**Optimization.** We optimize both networks jointly using AdamW with learning rate $10^{-4}$ and batch size 400. Training is run for 5,000 iterations on a single GPU.

**Edge recovery and ROC evaluation.** To quantify structure recovery, we assign each unordered pair of nodes $(i, j)$ an *edge score* $s_{ij} \in \mathbb{R}$ produced by the learned model (larger scores indicate stronger evidence of an edge). For our method, we obtain directional scores from masked conditional queries: for each target node $i$ we feed the encoder a one-hot target mask $m = e_i$ and read out the gate values $g_i(j)$ measuring how strongly node $j$ is selected to predict $i$. We then form an undirected score by symmetrization,

$$s_{ij} = \max\{g_i(j), g_j(i)\}, \qquad i < j.$$

Ground-truth edges are defined from the true precision matrix $\Theta^\star$ as

$$y_{ij} = \mathbb{I}\{|\Theta_{ij}^\star| > \varepsilon\}, \qquad i < j,$$

where $\varepsilon$ is a small numerical threshold (and we ignore diagonal entries). Given scores $\{s_{ij}\}_{i<j}$ and labels $\{y_{ij}\}_{i<j}$, we sweep a decision threshold $\tau$ over the score range and predict an edge whenever $s_{ij} \geq \tau$. For each $\tau$, we compute the true positive rate and false positive rate,

$$\text{TPR}(\tau) = \frac{\sum_{i<j} \mathbb{I}\{y_{ij} = 1, \, s_{ij} \geq \tau\}}{\sum_{i<j} \mathbb{I}\{y_{ij} = 1\}}, \qquad \text{FPR}(\tau) = \frac{\sum_{i<j} \mathbb{I}\{y_{ij} = 0, \, s_{ij} \geq \tau\}}{\sum_{i<j} \mathbb{I}\{y_{ij} = 0\}},$$

which traces out the ROC curve $\{(\text{FPR}(\tau), \text{TPR}(\tau))\}_\tau$; the AUC is computed as the area under this curve. This evaluation is threshold-free and measures how well the method ranks true edges above non-edges.

**Baseline (Graphical Lasso).** As a classical likelihood-based baseline, we fit a sparse Gaussian graphical model using the $\ell_1$-penalized log-likelihood (graphical lasso). The regularization strength (penalty parameter) is selected via cross-validation using `sklearn`'s `GraphicalLassoCV`, and the resulting estimated precision matrix $\widehat{\Theta}_{\text{gl}}$ is converted into edge scores using $s_{ij} = |\widehat{\Theta}_{\text{gl},ij}|$ for $i < j$. We then compute ROC and AUC using the same procedure and the same ground-truth labels $y_{ij}$.

**PC baseline.** We compare our method with the PC algorithm (Spirtes et al., 2000), a constraint-based causal discovery method that infers graph structure by testing conditional independencies. Let $X = (X_1, \ldots, X_d) \sim \mathcal{N}(0, \Sigma)$ with precision matrix $\Theta = \Sigma^{-1}$. PC performs a sequence of statistical tests of the null hypothesis $H_0 : X_i \perp\!\!\!\perp X_j \mid X_S$ using Fisher's Z test, which is based on the sample partial correlation $\hat{\rho}_{ij|S}$. For a given significance level $\alpha$, the algorithm returns a graph with adjacency matrix $\widehat{A}(\alpha) \in \{0, 1\}^{d \times d}$, which we interpret as a binary estimator of the conditional independence structure. The ground-truth adjacency is defined by $A_{ij}^\star = \mathbb{I}\{|\Theta_{ij}| > 0\}$, so that the true Markov blanket of $X_i$ is $\text{MB}(X_i) = \{j \neq i : \Theta_{ij} \neq 0\}$. To evaluate structure recovery, we sweep $\alpha$ over a grid and compute

$$\text{TPR}(\alpha) = \frac{\sum_{i<j} \mathbb{I}\{\widehat{A}_{ij}(\alpha) = 1 \wedge A_{ij}^\star = 1\}}{\sum_{i<j} \mathbb{I}\{A_{ij}^\star = 1\}}, \quad \text{FPR}(\alpha) = \frac{\sum_{i<j} \mathbb{I}\{\widehat{A}_{ij}(\alpha) = 1 \wedge A_{ij}^\star = 0\}}{\sum_{i<j} \mathbb{I}\{A_{ij}^\star = 0\}}.$$

This yields a ROC curve for each run. We repeat the experiment over multiple independent datasets and report the mean AUC with a $\pm 1$ standard deviation band, providing a direct comparison with our method for Markov blanket recovery.

## E. Additional Results for Learning Markov Blanket in Section 6.1

### E.1. Learning Structure of Graphical Model

In this experiment, we compare the learned neural gates with the true precision matrix using two classic Gaussian graphical models: 3rd-order Markov chain and the lattice model. The data is generated using the technique described in Section 6.1. We learn the graphical model structure using two encoder architectures: LSTM and CNN. See Section D for details.

The results are shown in Figure 11. It can be seen that the learned neural gates accurately reflect the sparsity pattern in the precision matrix. These graphical models are recovered using 2048 samples generated from a 50-dimensional 3rd order Markov Chain and a 64- and 225-dimensional Lattice structure introduced in Section 6.1. We set `VectorField`$(d, 256)$ which is consistent with our ROC experiments.

Notice that in the case of 15 x 15 lattice, the number of total potential edges in the graphical model is $225 \times 224/2 = 25,088$, much larger than the sample size 2048. Thus, inductive bias is almost necessary for recovering the graphical model.

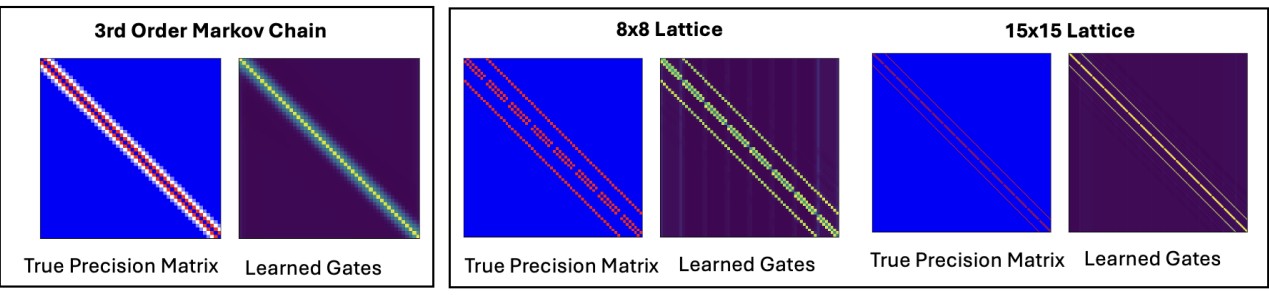

*Figure 11.* Comparison between true precision matrices and learned neural gates on two different graphical model structures. The non-zero gates align well with the non-zero entries of the true precision matrices.

# F. Implementation of Representation Learning Experiments in Section 6.2

This section describes the implementation details for the Shortcut problem experiments.

## F.1. Data augmentation and watermarking

### F.1.1. DATA AUGMENTATION

The data augmentation is done via the following pipeline:

1. **Random Resized Crop:** A region of the image is randomly cropped and resized to a fixed resolution of $32 \times 32$. The scale of the cropped area is sampled uniformly from the range $[0.8, 1.0]$ of the original image area.

2. **Random Horizontal Flip:** The cropped image is horizontally mirrored with a probability of $p = 0.5$, introducing reflection invariance.

### F.1.2. WATERMARKING VIA RANDOM COLOR INJECTION

In this strategy, we assign a random RGB vector to each image in the dataset, effectively "watermarking" the originally grayscale inputs with a unique color profile, thereby providing a shortcut for representation-learning algorithms.

### F.1.3. WATERMARKING VIA RANDOM PATCHING

In this strategy, we implement a patch-based watermarking scheme. We generate a unique, fixed $10 \times 10$ pixel RGB patch, $\mathbf{p}_i \in [0, 1]^{3 \times 10 \times 10}$, for every sample index $i$ in the dataset. These patches are sampled from a uniform distribution $\mathcal{U}(0, 1)$. This effectively assigns a high-frequency "watermark" to each image, providing a localized shortcut for the representation-learning algorithms.

## F.2. Zero-flow Network Architecture

In this section, we describe the neural network architecture used in the experiments. An illustration of the network can be found in Figure 12. Note that there is a minor difference in the network inputs between training and inference. In the following text, we assume the inference time setting.

Let $X, Y \in \mathbb{R}^{3 \times H \times W}$ denote the input images with height $H$ and width $W$, and let $f(Y) \in \mathbb{R}^L$ denote the latent vector of dimension $L$.

### F.2.1. ENCODER

For both SimCLR and zero-flow, we use the same encoder architecture to ensure fair comparison. The encoder, denoted as $E(\cdot)$, processes the input image $Y$ through a series of convolutional layers followed by a dense projection.

1. **Feature Extraction:** The input $Y$ is passed through two convolutional blocks. Each block consists of a convolution

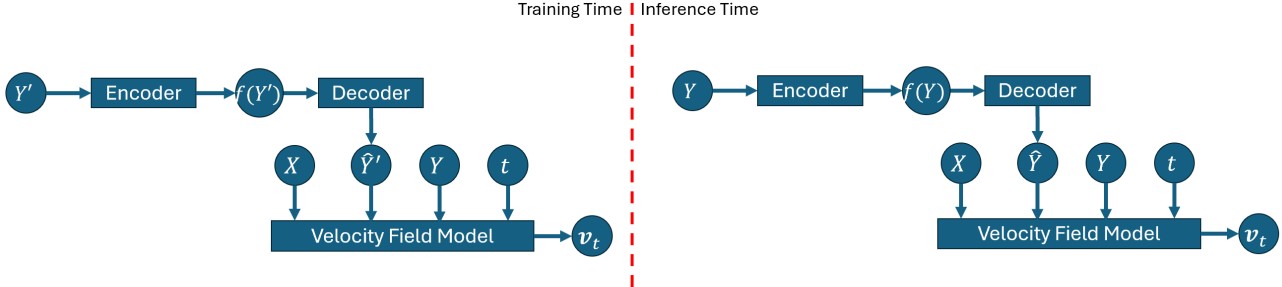

*Figure 12.* The neural network architecture for the representation learning Section 6.2.

with a $3 \times 3$ kernel, stride 1, and padding 1, followed by a ReLU activation. The channel dimensions increase from 3 (RGB) to 16.

2. **Channel Compression:** A third convolutional layer maps the 16 feature channels down to a single channel ($C = 1$), maintaining the spatial resolution $H \times W$.

3. **Latent Projection:** The resulting $1 \times H \times W$ feature map is flattened into a vector of size $HW$. A fully connected (linear) layer projects this vector to the final latent dimension $L$.

Formally, the encoder operation can be summarized as:

$$\mathbf{z} = \text{Linear}_{HW \to L}(\text{Flatten}(\text{Conv}_{16 \to 1}(\dots))) \tag{37}$$

### F.2.2. DECODER

The decoder, $D(\cdot)$, utilizes an upsampling strategy to reconstruct the image $\hat{Y}$ from $f(Y)$.

**Upsampling Blocks:** The feature map is progressively upsampled to the target resolution $H \times W$. Each upsampling block consists of:

- **Transposed Convolution:** A kernel size of $4 \times 4$, stride 2, and padding 1 is used to exactly double the spatial dimensions.

- **Batch Normalization & ReLU:** Applied after every transposed convolution except the output layer.

- **Channel Reduction:** The channel depth is halved at each layer (e.g., $64 \to 32 \to 16$) until the final layer.

The final layer maps the features to 3 channels followed by a Sigmoid activation function to constrain the output pixel values to $[0, 1]$.

$$\hat{Y} = \sigma(\text{ConvTranspose}(\dots)) \tag{38}$$

### F.2.3. VELOCITY FIELD ARCHITECTURE

The velocity field network, denoted as $V(\cdot)$, models the time derivative of the system state, $\frac{d\mathbf{x}}{dt}$.

Unlike the encoder, this network operates on a spatially dense representation. To predict the update for the current state $X$, the network conditions on three auxiliary inputs: $Y$, the decoded representation $\hat{Y}$, and the scalar time $t$.

**Input Alignment and Concatenation** Since the inputs possess varying dimensions (scalars, vectors, and image tensors), they must first be aligned to a common spatial resolution $H \times W$ before processing. The alignment process is defined as follows:

1. $X_t$**:** The input state is reshaped to the standard image tensor format $\mathbf{X} \in \mathbb{R}^{3 \times H \times W}$.

2. $Y$: The vector is reshaped to match the spatial dimensions, resulting in a tensor $\mathbf{Y} \in \mathbb{R}^{3 \times H \times W}$.

3. **Decoded** ($\hat{Y}$) The latent vector reshaped to match the spatial dimensions, resulting a tensor: $\hat{\mathbf{Y}} = D(f(Y)) \in \mathbb{R}^{3 \times H \times W}$.

4. **Time Embedding** ($t$): The scalar time input is spatially expanded across the height and width, forming a constant feature map $\mathbf{T} \in \mathbb{R}^{1 \times H \times W}$.

These four tensors are concatenated along the channel dimension to form the input tensor $\mathbf{I}_{in}$:

$$\mathbf{I}_{in} = \text{Concat}(\mathbf{x}, \mathbf{Y}, \hat{\mathbf{Y}}, \mathbf{T}) \in \mathbb{R}^{(3+3+3+1) \times H \times W} = \mathbb{R}^{10 \times H \times W} \tag{39}$$

**Convolutional Processing**   The concatenated input $\mathbf{I}_{in}$ is processed by a shallow Convolutional Neural Network (CNN) to compute the gradient field. The network consists of three convolutional layers with $3 \times 3$ kernels, stride 1, and padding 1 to preserve spatial resolution:

- **Layer 1:** Maps 10 input channels to 16 hidden channels, followed by a ReLU activation.

- **Layer 2:** Maps 16 channels to 16 channels, followed by a ReLU activation.

- **Layer 3 (Output):** Maps 16 channels to 3 output channels (corresponding to RGB velocity). No activation function is applied, allowing the vector field to span the full real range $(-\infty, \infty)$.

The output is flattened to a vector of size $3HW$, representing the derivative at every pixel location.

### F.2.4. TRAINING

We train both SimCLR and the Zero-flow encoders with 5000 iterations on the training set, using Adam optimizer with a 10e-4 learning rate.

### F.3. MAE Model Architecture

The Masked Autoencoder (MAE, (He et al., 2022)) leverages Transformer architectures (Vaswani et al., 2017) to capture global contextual dependencies via self-attention, which fundamentally differs from the previous two convolutional-based models. To ensure a fair comparison without introducing excessive model capacity, we adopt a streamlined Transformer-based architecture for both the encoder and the decoder.

### F.3.1. RANDOM MASKING

To enable masked image modeling, a random masking operation is applied to the patch token sequence prior to encoding. The masking procedure is summarized as follows:

1. **Patch Tokenization:** The input image $\mathbf{x} \in \mathbb{R}^{C \times H \times W}$ is partitioned into non-overlapping $4 \times 4$ patches and linearly projected into a sequence of patch tokens $\mathbf{X} \in \mathbb{R}^{N \times D_e}$, where $D_e = 192$ and $N = HW/16$.

2. **Random Masking:** A random subset of patch tokens is removed according to a predefined mask ratio $r = 0.75$, yielding a set of visible tokens $\mathbf{X}_{\text{vis}} \in \mathbb{R}^{N_{\text{vis}} \times D_e}$, where $N_{\text{vis}} = (1 - r)N$.

3. **Encoder Input:** The visible patch tokens are concatenated with a learnable class token and passed to the encoder.

Formally, the masking operation can be written as

$$\mathbf{X}_{\text{vis}} = \text{Select}_{N_{\text{vis}}}(\text{PatchEmbed}_{4 \times 4}(\mathbf{x})), \tag{40}$$

where $\text{Select}_{N_{\text{vis}}}(\cdot)$ denotes random selection of $N_{\text{vis}}$ patch tokens.

### F.3.2. ENCODER

The encoder adopts a tiny Vision Transformer (ViT-Ti, (Touvron et al., 2021)) architecture and maps the partially observed patch tokens to latent representations. Given the visible patch tokens produced by the random masking process, the encoding procedure is defined as follows:

1. **Positional Encoding:** Positional embeddings are added to each visible patch token to encode spatial information. These embeddings are implemented as learnable absolute positional embeddings and are jointly optimized with the encoder parameters.

2. **Class Token Injection:** A learnable class token is augmented with its corresponding positional embedding and prepended to the sequence of visible patch tokens, serving as a global image-level representation.

3. **Transformer Encoding:** The resulting token sequence is processed by a Transformer encoder consisting of 12 Transformer blocks, each equipped with 3 self-attention heads. Layer normalization is applied to the encoder output.

Formally, the encoder operation can be summarized as

$$\mathbf{H}_{\text{enc}} = \text{LN}(\text{Transformer}_{12,3}(\text{Concat}(\mathbf{c} + \mathbf{p}_{\text{cls}}, \mathbf{X}_{\text{vis}} + \mathbf{P}_{\text{vis}}))) \in \mathbb{R}^{(1+N_{\text{vis}}) \times D_e}, \tag{41}$$

where $\mathbf{X}_{\text{vis}}$ denotes the visible patch tokens, $\mathbf{P}_{\text{vis}}$ denotes their positional embeddings, $\mathbf{c}$ denotes the learnable class token, and $\mathbf{p}_{\text{cls}}$ denotes its positional embedding.

### F.3.3. DECODER

The decoder maps the latent representations produced by the encoder to patch-level pixel predictions using a lightweight Transformer architecture. Let $\mathbf{H}_{\text{enc}} \in \mathbb{R}^{(1+N_{\text{vis}}) \times D_e}$ denote the encoder output including the class token. The decoding procedure is defined as follows:

1. **Class Token Removal and Latent Projection:** The class token is removed from the encoder output, yielding $\mathbf{H}_{\text{vis}} \in \mathbb{R}^{N_{\text{vis}} \times D_e}$, which contains only the patch-level latent representations. These representations are then linearly projected from the encoder embedding dimension $D_e$ to a lower-dimensional decoder embedding space with dimension $D_d = 128$.

2. **Mask Token Injection:** Learnable mask tokens are appended to the projected latent representations to form a complete token sequence for decoding, corresponding to both visible and masked patches.

3. **Transformer Decoding and Patch Prediction:** Learnable absolute positional embeddings are added to the restored token sequence to encode spatial information. These embeddings are optimized jointly with the decoder parameters. The position-augmented tokens are then processed by a Transformer decoder consisting of 2 Transformer blocks with 4 self-attention heads each. The decoder output is passed through layer normalization and a linear prediction head that projects the decoder embeddings to patch-level pixel values of dimension $4 \times 4 \times C$.

Formally, the decoder operation can be summarized as

$$\hat{\mathbf{X}} = \text{Linear}_{D_d \to 4 \times 4 \times C}\big(\text{LN}\big(\text{Transformer}_{2,4}\big(\text{Concat}\big(\text{Proj}_{D_e \to D_d}(\mathbf{H}_{\text{vis}}), \mathbf{m}\big) + \mathbf{P}\big)\big)\big), \tag{42}$$

where $\mathbf{H}_{\text{vis}} \in \mathbb{R}^{N_{\text{vis}} \times D_e}$ denotes the encoder outputs after removing the class token, $\mathbf{m}$ denotes the learnable mask tokens, and $\mathbf{P}$ denotes the decoder positional embeddings.

The reconstruction loss is computed only on the masked patches, encouraging the encoder to infer missing content from global context.

$$\mathcal{L}_{\text{MAE}}(f) = \mathbb{E}\big[\|\mathbf{X}_{-\mathbf{m}} - \text{Decoder}(\text{Encoder}(\mathbf{X}_{\mathbf{m}}))\|^2\big],$$

After pretraining, the decoder is discarded, and only the pretrained encoder is retained for downstream evaluation.

F.3.4. LINEAR PROBING PROTOCOL

To evaluate the quality of representations learned by the encoder, we adopt a linear probing protocol consistent with the standard evaluation setting used in MAE. During linear probing, the encoder is used as a fixed feature extractor. The procedure is defined as follows:

1. **Frozen Encoder with Global Average Pooling:** A deep copy of the pretrained encoder is used to prevent any modification of the original model. All encoder parameters are frozen. To obtain a single image-level representation, global average pooling is applied to the encoder outputs: the class token is discarded, and the remaining patch token representations are averaged across the spatial dimension, followed by layer normalization.

2. **Linear Classifier Training:** A linear classifier $LC_\theta$ is trained on top of the frozen encoder representations using labeled training data by minimizing the cross-entropy loss.

3. **Evaluation on Clean Test Data:** The trained linear classifier is evaluated on a clean and normalized test set that is independent of the training data. Classification accuracy on this test set is reported as the linear probing performance.

Formally, given an input image $\mathbf{x}$ and its label $y \in \{1, \ldots, K\}$, let $\mathbf{z} = \mathrm{LN}\left(\frac{1}{N} \sum_{i=1}^{N} \mathrm{Encoder}(\mathbf{x})_i\right)$ denote the pooled encoder representation, where $\mathrm{Encoder}(\mathbf{x})_i$ is the representation of the $i$-th visible patch token and $N$ is the number of visible patches. The linear classifier $LC_\theta$ is trained by minimizing the cross-entropy loss

$$\mathcal{L}_{\mathrm{CE}}(\mathbf{x}, y) = -\log \mathrm{Softmax}(LC_\theta(\mathbf{z}))_y. \tag{43}$$

The predicted label is then obtained as $\hat{y} = \arg\max_{k \in \{1,\ldots,K\}} LC_\theta(\mathbf{z})_k$, where $K$ denotes the number of classes.

The linear probing accuracy is computed as

$$\mathrm{Acc} = \frac{1}{|\mathcal{D}_{\mathrm{test}}|} \sum_{(\mathbf{x}, y) \in \mathcal{D}_{\mathrm{test}}} \mathbb{I}(\hat{y} = y), \tag{44}$$

where $\mathcal{D}_{\mathrm{test}}$ denotes the clean test dataset and $\mathbb{I}(\cdot)$ is the indicator function.

To reduce the effect of randomness, the entire linear probing procedure is repeated five times with different random seeds. The final reported performance is the mean linear probing accuracy over 5 runs.

F.3.5. MAE TRAINING

We train the MAE model for 100 epochs on the training set using the AdamW optimizer, with a learning rate of 10e-4 and a weight decay of $0.05$.

