# OpenReview forum: "Zero-Flow Encoders"
_ICML.cc/2026/Conference — ICML 2026 regular_

### Official Review · Reviewer_MiHo · 2026-03-10

**Soundness:** 2
**Presentation:** 3
**Significance:** 2
**Originality:** 4
**Overall Recommendation:** 4
**Confidence:** 4

**Summary:**

This work presents a new measure for independence: two RVs are independent iff the optimal flow velocity is zero at t=0.5. They evaluate their method on one toy sequence dataset and one toy image dataset.

**Compliance With Llm Reviewing Policy:**

Affirmed.

**Final Justification:**

The rebuttal fully addresses my concern. I'm raising my score.

**Key Questions For Authors:**

- Why is zero-flow better than the simple reconstruction criterion for testing independence?
- In Sec. 6.2, why do we expect the model not to take this "shortcut", and in particular, why does zero-flow not do this despite it being an optimal strategy to find the Markov blanket?

**Limitations:**

yes

**Strengths And Weaknesses:**

Strength

- The idea of using optimal rectified flow for testing independence is simple and new.

Weakness
- A straightforward way to test independence is the reconstruction test, which can lower-bound the mutual information. Intuitively, if a sufficiently powerful neural decoder cannot predict X from f(Y), f(Y) doesn't have any information about X. I don't see why the zero-flow criterion should be better than this simple ubiquitous method.
- In Sec. 6.2, the authors claim that the proposed method does not take shortcuts. But why? If the injected 10x10 patch in the top-left corner contains sufficient information to predict the target, it is optimal to just take this feature. I don't see why we expect the model not to take this "shortcut", and in particular, why zero-flow does not do this, despite this being an optimal strategy to find the Markov blanket.

---

> ### Author Rebuttal · Authors · 2026-03-30
>
> > Why is zero-flow better than the simple reconstruction criterion for testing independence?
>
> Thank you for the comment!
>
> Although we can indeed understand sufficiency in terms of reconstructability, **performing a distribution-free reconstruction test is generally intractable**.
>
> Simple reconstruction criteria, such as a least-squares loss, **only shows how well $f(Y)$ recovers $\mathbb{E}[X|Y]$, not the  distribution $X|Y$**. This loss also implies $X|Y$ and $X|f(Y)$ are Gaussian, thus breaking the promise of distribution-free test. To measure true reconstruction quality without restrictive modeling assumptions, one must estimate the full likelihoods $ p (X| Y) $ and $ p (X|f(Y)) $ and compare them. While conventional non-parametric methods (such as kernel estimators) can theoretically achieve this, they suffer severely from the curse of dimensionality.  If $X$ and $Y$ are in a high-dimensional space, estimating a high-dimensional likelihood estimation is difficult and is extremely time-consuming (the KCI mentioned in Section 6.1 failed for exactly this reason.)
>
> In short, testing conditional independence via reconstruction error is typically intractable due to estimating likelihood in a high-dimensional space. **This is where the Zero-Flow criterion shines**: it bypasses the need for intractable density or likelihood estimation. By evaluating the vector field at the midpoint, Zero-Flow directly enforces full distributional comparison through flow dynamics, offering a highly scalable, distribution-free test for conditional independence and enabling various high-dimensional, large-scale downstream applications.
>
> >In Sec. 6.2, the authors claim that the proposed method does not take shortcuts. But why?
>
> To clarify, our method learns **both the shortcut and non-shortcut** features. However, when a shortcut exists, SimCLR learns shortcut features **almost exclusively**, without recognizing any other meaningful features.
>
> This is a fundamental limitation of SimCLR, which frames representation learning as an instance-wise classification problem. The contrastive loss acts as a classification loss, which decreases as long as the current encoded sample ($f(X_i)$) is easily separated from the negatives ($f(X_{-i})$). If there exists a shortcut (e.g., a watermark) that readily distinguishes the positive sample from the negatives, the classification loss quickly saturates. Once this shortcut is learned, the model has no incentive to learn additional, harder-to-extract features because the loss cannot be meaningfully reduced any further.
>
> In contrast, our zero-flow criterion is not a classification loss; **it is a distribution matching objective that demands full sufficiency**. A shortcut feature can not easily saturate the loss. Therefore, the model is forced to continue learning until all sufficient features—both the shortcuts and the underlying semantics—are fully identified.
>
> Here we provide a visual example (https://i.ibb.co/GNXdRgk/Screenshot-2026-03-31-at-12-39-05.png), in which we trained all encoders to reconstruct an unseen, watermarked image. It can be seen that both encoders pick up the watermark information (the box in the top-left corner). However, SimCLR only picks up the watermark information and fails to reconstruct any meaningful visual patterns of the original image. This can also be validated by the results in our paper (Figure 6): our representation is sensitive to both the shape and color of clothes.
>
> > larger scale experiments
>
> **Note that we now demonstrate this phenomenon on larger-scale datasets (including ImageNet-1K)**, and the results are presented in our response to reviewer 3uP5. Not only does the shortcut severely affect the linear probe accuracy, but it also leads to large reconstruction errors. In comparison, our method (and MAE) **performs robustly on the watermarked datasets**.

---

> > ### Author Rebuttal · Reviewer_MiHo · 2026-04-04
> >
> > My concerns are fully resolved. I'm raising my score.

---

> > > ### Author Response · Authors · 2026-04-04
> > >
> > > Thank you for the review comments and reply to our rebuttal!! We will add clarifications of these points in our revision.

---

### Official Review · Reviewer_9KUd · 2026-03-13

**Soundness:** 3
**Presentation:** 3
**Significance:** 3
**Originality:** 3
**Overall Recommendation:** 4
**Confidence:** 4

**Summary:**

The authors propose zero-encoders by identifying a zero-flow phenomenon in flow matching frameworks. Showing that this criterion is zero at $t=0.5$, the authors unveil the possibility of learning representations. Specifically, the new metric, “zero-flow”, is proved to show the conditional independence of having $f$ as a meaningful encoder implies $f$ learns the summarized statistics to encode the data. Then to enforce the encoder with this property, the authors propose zero-flow loss. Empirically, they identify that this framework can be used to extract the Markov blanket for graphical models, as well as learning visual representations for MINIST and CIFAR.

**Compliance With Llm Reviewing Policy:**

Affirmed.

**Final Justification:**

The paper presents interesting and novel idea. However, the empirical evidence regarding the scalability and generation quality is not clear. Still, I like to support new idea and therefore maintain my positive boardline score.

**Key Questions For Authors:**

1. I am wondering what are the limits for scaling up to large systems and high-dimensional datasets. Especially, training the model on the pixel space would be unaffordable using flow-matching on datasets like ImageNet and CoCo, so a practical solution is to leverage a pre-trained encoder. I am not sure what's the performance for this proposed method to be applied to such real cases.

2. Additionally, how's the generative results? I understand that the paper is targeting representation learning, however, if the reconstruction accuracy is lower by having this additional regularization, one could imagine that the method will be difficult to train and scale up.

Therefore, I would really want to see a detailed discussion and results for addressing these aspects. Thank you!

**Strengths And Weaknesses:**

### Strength:

1.The idea of this “zero-flow” criterion is novel. And the proof is sound.

2.The empirical study has verified the proposed idea to some extent.

### Weakness:

It’s not clear what’s the limitations of having the zero-flow regularization, in particular, one could easily imagine that forcing the velocity to be zero could not scale to real applications.

---

> ### Author Rebuttal · Authors · 2026-03-30
>
> >what are the limits for scaling up to large systems and high-dimensional datasets... so a practical solution is to leverage a pre-trained encoder.
>
> We thank the reviewer for raising this important practical consideration. We completely agree that performing joint flow-matching directly on high-resolution pixel spaces is computationally prohibitive for datasets like ImageNet or COCO.
>
> We would like to note that the Rectified flow and the neural vector field models scale well to moderately high-dimensional datasets. For example, it has been trained natively on Imagenet (256 x 256) [1] and CelebA HQ (512 x 512) [2].
>
> Furthermore, we would like to highlight that **our zero-flow encoder is fully compatible with latent space flow matching (as suggested)**.
> For high-resolution real-world image datasets, modeling raw RGB pixels is often unnecessary. The standard practice of using a lightweight encoder (e.g., a VAE or a Vision Transformer) projects the high-dimensional image into a lower-dimensional latent space and then performs flow matching.
>
> Suppose we have a pretrained encoder for $X$,  $\phi: \mathbb{R}^{d} \to \mathbb{R}^b$, then we can run a zero-flow encoder in this latent space, optimizing
> \begin{align}
>     \min_{f, v} ||\phi(X) - \phi(X') - v(\phi(X), f(Y), Y')||^2 + ||v(\phi(X), f(Y), Y)||^2.
> \end{align}
> If $\phi(X)$ is sufficient for predicting $X$ (i.e., $\phi$ is an expressive encoder), then the conditional independence $\phi(X) \perp Y |f(Y)$ strictly implies $X \perp Y |f(Y)$.
>
> With our Zero-Flow criterion and theoretical guarantees (Theorem 4) holding perfectly in the latent space, there is no fundamental mathematical or architectural bottleneck preventing our Zero-Flow encoder from scaling to high-dimensional datasets using the suggested pretrained encoders. We will highlight this exciting direction in our revision!
>
> 1. Liu et al., Flow Straight and Fast: Learning to Generate and Transfer Data with Rectified Flow, ICLR2023
>
> 2. Esser et al., Scaling Rectified Flow Transformers for High-Resolution Image Synthesis, CVPR2024
>
> >Additionally, how's the generative results?
>
> Thanks for this excellent suggestion!
>
> To address this question, we have tested reconstruction errors for all methods in our additional experiments with larger-scale datasets suggested by reviewer 3uP5. After training each encoder, we train a decoder with a 2D transposed convolutional architecture by minimizing the least-squares reconstruction loss and report the test reconstruction error.
>
> | Dataset | Method | Reconstruction MSE (clean) | Reconstruction MSE (watermarked) |
> |---|---|---|---|
> | STL-10 | Ours | 0.0259 | 0.0256 |
> | STL-10 | SimCLR | 0.0464 | 0.0674 |
> | STL-10 | MAE | **0.0246** | **0.0245** |
> | TinyImageNet | Ours | **0.0285** | **0.0290** |
> | TinyImageNet | SimCLR | 0.0491 | 0.0758 |
> | TinyImageNet | MAE | 0.0312 | 0.0305 |
> | ImageNet | Ours | **0.0164** | **0.0161** |
> | ImageNet | SimCLR | 0.0299 | 0.0660 |
> | ImageNet | MAE | 0.0202 | 0.0177 |
>
> The experiments show that our method, despite not being trained with a reconstruction loss, **achieves the lowest reconstruction error on two out of three tested datasets** and is also robust to watermarking. Both our method and SimCLR use the same CNN-based encoder, whereas MAE uses a vision transformer (ViT-Tiny).
>
> We also provide a visualization of ImageNet here (https://i.ibb.co/WptpQn8D/Screenshot-2026-03-31-at-10-38-33.png), where we can see that SimCLR's reconstruction is visibly worse than our method, and completely fails after watermark is introduced.

---

> > ### Author Rebuttal · Reviewer_9KUd · 2026-04-02
> >
> > I appreciate the authors' response. Though I am willing to support the paper's acceptance, I need to admit that my concerns are not fully resolved, especially regarding the scaling up. The additional visualization does look interesting, but I am not sure why we want to compare to SimCLR, which is a representation learning framework, to show the generation performance.
> >
> > Still I like the idea. I will maintain my positive evaluation.

---

> > > ### Author Response · Authors · 2026-04-03
> > >
> > > We thank the reviewer for maintaining a positive rating of our paper!
> > >
> > > Sorry that we missed one of your earlier comments on the scalability of our method:
> > >
> > > >It’s not clear what’s the limitations of having the zero-flow regularization, in particular, one could easily imagine that forcing the velocity to be zero could not scale to real applications.
> > >
> > > We would like to clarify that **forcing the velocity to zero incurs no additional computational cost compared to an ordinary least squares loss**. Solving the zero-flow loss in equation 8 has exactly the same computational complexity as solving a regular least-squares objective. The term that enforces the velocity to be zero, i.e., the regularization term $\int \omega(t) \mathbb{E}[||u_t||^2]\mathrm{d}t$ in equation 8 should scale at the same rate as the rectified flow loss, i.e., O(nd), where $n$ is the batch size and $d$ is the dimensionality of $X$.
> > >
> > > >but I am not sure why we want to compare to SimCLR, which is a representation learning framework, to show the generation performance.
> > >
> > > Our method is also a representation-learning method that leverages the same multi-view principle as SimCLR (see Section 5.1), for this reason, we feel SimCLR is a reasonable comparison.
> > >
> > > Note that in the table above (and in the paper), we also compared against MAE. We believe this is a strong baseline for the reconstruction task. MAE learns representations by reconstructing "artificial missing patches" in the images. Thus, it is naturally suited for reconstruction tasks.
> > >
> > > >However, if the reconstruction accuracy is lower by having this additional regularization, one could imagine that the method will be difficult to train and scale up.
> > >
> > > The results in the table above show that **our method achieves superior reconstruction accuracy on two of the three datasets** (and remains competitive on the other), indicating no performance degradation from enforcing zero-flow regularization.
> > >
> > > Here, we provide a visualization of the reconstruction task using both our method and MAE. (https://i.ibb.co/S7D0CCbf/Screenshot-2026-04-03-120817.png).
> > >
> > >
> > > ---
> > > Please do not hesitate to let us know if any concerns remain unaddressed or if there is any misunderstanding of your review comments!

---

### Official Review · Reviewer_3uP5 · 2026-03-13

**Soundness:** 2
**Presentation:** 2
**Significance:** 2
**Originality:** 3
**Overall Recommendation:** 3
**Confidence:** 4

**Summary:**

This paper introduces Zero-Flow Encoders, a representation learning framework that repurposes the velocity fields of rectified flows. The authors prove that the flow vanishes at the temporal midpoint if and only if the source and target distributions match, a property used to enforce conditional independence. This criterion is applied to extract non-parametric latent representations and learn amortized Markov blankets in graphical models.

**Compliance With Llm Reviewing Policy:**

Affirmed.

**Final Justification:**

The core contribution of this paper—bridging generative flow-matching with discriminative representation learning through the "zero-flow" criterion—is highly original and provides a fresh perspective on self-supervised learning. In the rebuttal, the authors effectively addressed the robustness of the method against "shortcut" patterns (like watermarks), which is a notable advantage over contrastive baselines like SimCLR.
​Admittedly, some weaknesses persist: the ImageNet-1K experiments were conducted at a low resolution (32 \times 32), which does not fully showcase the method's potential on high-fidelity data. Additionally, the computational overhead of training an auxiliary velocity field remains a practical concern. However, the authors' final clarification regarding the baseline performance—explaining that the near-zero accuracy of SimCLR was specifically due to the watermark-induced shortcut rather than a flawed setup—partially alleviates my concerns regarding the experimental integrity.
​Given the conceptual novelty and the theoretical elegance of the zero-flow encoder, I believe the paper’s contribution to the field outweighs its empirical limitations. While I maintain my original score to reflect the need for more standard large-scale benchmarking in future versions, I do not oppose the acceptance of this paper if the Area Chair finds the theoretical novelty sufficient.

**Key Questions For Authors:**

- Could the authors provide results on a larger benchmark, such as ImageNet-1K, to demonstrate how the Zero-Flow Encoder performs when exposed to the high-data regimes where baselines like MAE typically excel?
- What is the specific quantitative increase in training time and memory consumption introduced by the velocity field model relative to the standard encoder-only baselines?
- Are there potential strategies or transformations that could extend the zero-flow criterion to handle discrete data structures while maintaining the current theoretical guarantees?

**Limitations:**

Yes

**Strengths And Weaknesses:**

Strengths:

- Originality: The work provides a highly novel bridge between generative flow-matching and discriminative feature extraction.
- Robustness to Shortcuts: The method shows a significant ability to ignore superficial data patterns, such as watermarks, which often degrade the performance of contrastive baselines.

Weaknesses:

- Limited Experimental Scale: Evaluations are primarily conducted on smaller datasets like CIFAR-10 and MNIST. Because self-supervised methods such as SimCLR and MAE are known to be data-hungry and typically evaluated at the ImageNet-1K scale, the lack of large-scale experiments makes it difficult to assess the method's true scalability.
- Modeling Overhead: The approach necessitates training an auxiliary velocity field model in addition to the encoder, increasing the overall computational and architectural complexity.
- Lack of Discrete Support: The framework is inherently designed for continuous-valued distributions, leaving it inapplicable to datasets with discrete or categorical variables.

---

> ### Author Rebuttal · Authors · 2026-03-30
>
> > require tests on large-scale datasets.
>
> We completely agree with the reviewer that representation learning is often evaluated on larger-scale datasets.
>
> To address this, we have extended our representation learning experiments to include larger datasets, specifically STL-10 (96 x 96), Tiny ImageNet (64 x 64), and ImageNet1K (32 x 32). To further validate the learned representations, we evaluated image reconstruction performance by training a CNN decoder with a least-squares loss and measuring reconstruction errors on the watermarked testing sets.
>
> | Dataset | Method | Accuracy | Reconstruction MSE |
> |---|---|---|---|
> | STL-10 | Ours | 52.21% (+2.41) | 0.0256 (-0.0003)|
> | STL-10 | SimCLR | 10.00% (-60.23) | 0.0674 (+0.0210) |
> | STL-10 | MAE | **56.14%** (+0.65) | **0.0245 (-0.0001)** |
> | | | | |
> | TinyImageNet | Ours | 16.60% (+0.50) | **0.0290 (+0.0005)** |
> | TinyImageNet | SimCLR | 0.50% (-30.63) | 0.0758 (+0.0267) |
> | TinyImageNet | MAE | **18.75%** (-0.24) | 0.0305 (-0.0007) |
> | | | | |
> | ImageNet - 1K | Ours | 7.01% (+0.82) | **0.0161 (-0.0003)** |
> | ImageNet - 1K | SimCLR | 0.10% (-15.18) | 0.0660 (-0.0361) |
> | ImageNet - 1K | MAE | **8.1%** (-0.2) | 0.0177 (-0.0025) |
>
> **In brackets, we report the change relative to the unwatermarked dataset.**
>
> As the results demonstrate, SimCLR continues to suffer from a severe shortcut problem even on these larger datasets: after introducing the watermark patch, SimCLR's testing accuracy drops by 58\% on STL-10, 30\% on Tiny ImageNet, and 15\% on Imagenet.
>
> Interestingly, although neither SimCLR nor our method is explicitly trained for data reconstruction, **our encoder achieves a much lower reconstruction error than SimCLR's, both before and after watermarking**. Furthermore, once the watermarks are applied, SimCLR's reconstruction error spikes and aligns with the performance degradation we observed in the linear probing accuracy.
>
> Notably, our method achieves **the lowest reconstruction error on two of three tested datasets** and is only slightly behind MAE on STL10 dataset.
>
> > Are there potential strategies or transformations that could extend the zero-flow criterion to handle discrete data structures
>
> One simple strategy for handling discrete data is to embed it in a continuous space, then apply our method.
> For example, if $X, X' \in \mathcal{X}$ are discrete data and $\phi: \mathcal{X} \to \mathbb{R}^b$ is a feature map, we can run the rectified flow on the extracted continuous features $\phi(X) \in \mathbb{R}^b, \phi(X')\in \mathbb{R}^b$.
>
> This simple strategy highlights a key advantage of our method: the zero-flow encoder is **fully compatible with latent-space flow matching**. If there exists an effective feature map $\phi\in \mathbb{R}^b$, we can bypass the original samples and perform zero-flow encoding in the feature space, easily handling data like language or graphs.
>
> It is also possible to apply our method directly to discrete datasets. Inspired by your question, we have derived a zero-flux criterion for a one-dimensional discrete random variable, which serves as a direct analogue to our zero-flow criterion.
> Consider an interpolation process $X_t$ between two independent random variables $X, X'$ [1].
> Define the probability flux as the infinitesimal jump probability $F_t(a,b) := \lim_{h\to 0} \frac{P(X_t = a, X_{t+h} = b)}{h}$, and the net flux is defined as $J_t(a,b) = F_t(a,b) - F_t(b, a)$. The zero-flux criterion states, $J_t(a,b) = 0 \iff X \overset{d}{=} X'$. The authors are happy to provide a detailed proof if required.
>
> 1. Gat et al., Discrete Flow Matching, NeurIPS2024.
>
> > increase in training time and memory consumption?
>
> Overall, training the additional velocity field increases runtime by 4x to 5x and peak memory by roughly 2x to 2.5x compared to the standard SimCLR baseline (which does not utilize the auxiliary vector field). However, we should contextualize this by comparing it against MAE, another method that learns complex reconstructive mappings.
>
> | Method | CIFAR-10 (32×32) | TinyImageNet (64×64) | STL-10 (96×96) |
> |---|---|---|---|
> | SimCLR | 0.07 (1.3GB) | 0.19 (4.8GB) | 0.38 (10.6 GB) |
> | MAE (Tiny-ViT) | 4.44 (5.5GB) | 4.51 (5.6GB) | 4.32 (5.8GB) |
> | Ours | 0.30 (2.9GB) | 0.86 (11.6GB) | 1.42 (25.4 GB) |
>
> **Table: Training time (min/200 batches) and peak memory usage on CIFAR-10, STL-10, and TinyImageNet**
>
> Our method exhibits a clear time-memory tradeoff compared to MAE. MAE processes a much smaller fraction of image patches (typically 25%) through its encoder, inherently suppressing its memory footprint. Our CNN model processes at full resolution, which requires more memory, but it is much faster than the Transformer backbone.
>
> To sum up, while our velocity field does incur overhead relative to an encoder-only SimCLR, it remains **highly competitive in training speed compared to other reconstructive representation-learning methods**. Moreover, it is more robust to watermarks than the encoder-only SimCLR method.

---

> > ### Author Rebuttal · Reviewer_3uP5 · 2026-04-04
> >
> > ​I thank the authors for the additional experiments. However, I remain unconvinced for the following reasons. First, the experiments on ImageNet-1K were conducted at a downsampled resolution of 32x32, which is not the standard setting (e.g., 256x256) for evaluating modern representation learning or generative models. Results on such low resolutions do not sufficiently demonstrate the method's capability to handle complex, high-resolution visual structures. Second, the extremely poor performance of the baselines (e.g., SimCLR achieving near-zero accuracy) is highly unconventional and raises serious concerns about whether the experimental setup, hyperparameter tuning, or evaluation protocol is appropriate. Due to these fundamental flaws in the experimental setting, I will maintain my original score.

---

> > > ### Author Response · Authors · 2026-04-04
> > >
> > > Thanks for the reply!
> > >
> > > >the extremely poor performance of the baselines (e.g., SimCLR achieving near-zero accuracy) is highly unconventional and raises serious concerns about whether the experimental setup, hyperparameter tuning, or evaluation protocol is appropriate
> > >
> > > We would like to clarify that the accuracy we reported in the table is **on the watermarked dataset**. The low performance of SimCLR is caused by the shortcut issue we aim to demonstrate. On the original dataset, SimCLR achieved around **15% accuracy** (see our explanation after the table).
> > >
> > > In the reply to reviewer 9KUd, https://i.ibb.co/WptpQn8D/Screenshot-2026-03-31-at-10-38-33.png, we showed the reconstructed image using trained representations. We can see **when trained on the original dataset, the representations produced by SimCLR are still very respectable**.
> > >
> > > In summary, SimCLR's poor performance stems from the shortcut issue we aim to demonstrate and is **not** a fundamental flaw of our experimental setting.
> > >
> > > ---
> > > **Update**
> > >
> > > >  the experiments on ImageNet-1K were conducted at a downsampled resolution of 32x32, which is not the standard setting (e.g., 256x256)
> > >
> > > Reconstruction Errors (MSE) on the standard ImageNet dataset (all images rescaled to 256 x 256):
> > >
> > > | Method | Original | Watermark |
> > > | :--- | :--- | :--- |
> > > | Our method | 0.027247 | 0.027944 |
> > > | SimCLR | 0.049620 | 0.064526 |
> > >
> > > Our method still outperforms SimCLR in reconstruction quality and remains robust after introducing the watermark. Both SimCLR and our method use the same encoder model (three stacked CNN layers followed by an MLP projection layer).

---

### Decision · Program_Chairs · 2026-04-30

**Decision:**

Accept (regular)

**Comment:**

This paper introduces zero flow criteria, a theoretically grounded, distribution free method for testing conditioner independence. The review panel broadly recognizes the originality of the contribution. All three reviewer explicitly price the Novelty of the approach. The reviewer despite maintaining a weak reject score, explicitly stated that they do not oppose acceptance given the theoretical novelty. The remaining concern about large scale benchmarking is acknowledged, but ask the authors clarify this is a methodological paper introducing a novel theoretical framework rather than a systems paper target in SOTA benchmarks. I recommend accepting this paper on this strength of its theoretical contributions, the quality of its rebuttal and the positive consensus across the reviewers.